# ZSC-Eval: An Evaluation Toolkit and Benchmark for Multi-agent Zero-shot Coordination

**Xihuai Wang**,* **Shao Zhang**,* **Wenhao Zhang, Wentao Dong, Jingxiao Chen,**
**Ying Wen**,† **Weinan Zhang**†
Shanghai Jiao Tong University
{leoxhwang,shaozhang,ying.wen,wnzhang}@sjtu.edu.cn

## Abstract

Zero-shot coordination (ZSC) is a new cooperative multi-agent reinforcement learning (MARL) challenge that aims to train an ego agent to work with diverse, unseen partners during deployment. The significant difference between the deployment-time partners' distribution and the training partners' distribution determined by the training algorithm makes ZSC a unique out-of-distribution (OOD) generalization challenge. The potential distribution gap between evaluation and deployment-time partners leads to inadequate evaluation, which is exacerbated by the lack of appropriate evaluation metrics. In this paper, we present **ZSC-Eval**, the first evaluation toolkit and benchmark for ZSC algorithms. ZSC-Eval consists of: 1) Generation of evaluation partner candidates through behavior-preferring rewards to approximate deployment-time partners' distribution; 2) Selection of evaluation partners by Best-Response Diversity (BR-Div); 3) Measurement of generalization performance with various evaluation partners via the Best-Response Proximity (BR-Prox) metric. We use ZSC-Eval to benchmark ZSC algorithms in Overcooked and Google Research Football environments and get novel empirical findings. We also conduct a human experiment of current ZSC algorithms to verify the ZSC-Eval's consistency with human evaluation. ZSC-Eval is now available at https://github.com/sjtu-marl/ZSC-Eval.

## 1 Introduction

Building agents that can interact and collaborate with others without prior coordination in various scenarios is a crucial challenge of cooperative AI [50, 61, 9, 14]. One aspect of this challenge, known as Zero-shot coordination (ZSC) in cooperative multi-agent reinforcement learning (MARL) [55, 56, 63] involves developing an agent that learns coordination skills with a limited set of training partners and generalizes them to unseen partners during deployment [13, 34]. The distribution of training partners is determined by training algorithms, while deployment-time partners are determined by deployment requirements [16], making ZSC an out-of-distribution (OOD) generalization problem. ZSC capability evaluation requires specific methods, such as partners that meet deployment-time distributions and metrics that focus on generalization performance and not only task performance [25].

Current ZSC evaluation methods still face challenges. The distribution gap between evaluation and deployment-time partners is crucial. Human proxy agents might not fully mimic human behaviors [60], and generating evaluation and training partners using identical methods [51, 27] results in similar distributions, compromising the reliability of evaluation results. Cross-play evaluations among

---

*Both authors contribute equally to this work.
†Correspondence Authors

38th Conference on Neural Information Processing Systems (NeurIPS 2024) Track on Datasets and Benchmarks.

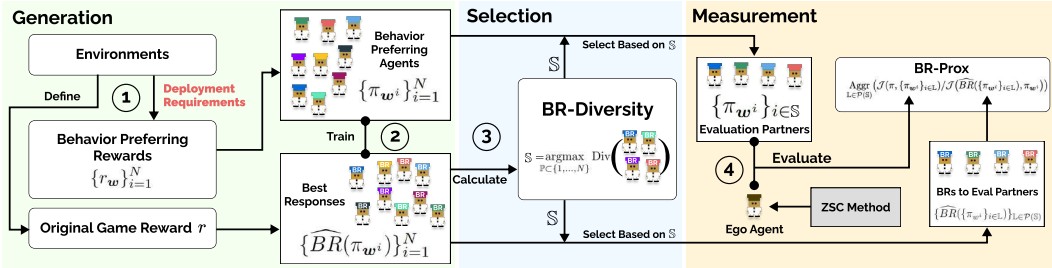

Figure 1: **ZSC-Eval.** 1) Generation: generating behavior-preferring agents and their best responses; 2) Selection: selecting evaluation partners by maximizing Best Response Diversity; 3) Measurement: evaluating the ego agent with the evaluation partners and computing Best Response Proximity.

trained ZSC agents [57, 23] risk unfair comparisons due to overlaps between training and evaluation partners. Some evaluation methods are inconvenient to implement, e.g., human proxy agents require human data for training. Moreover, using mean episode returns as the ZSC capability metric restricts evaluation to task performance, ignoring generalization performance like the generalization gap [16]. The mean episode returns also ignore different and unbalanced cooperation capabilities of evaluation partners [64]. Therefore, the community urgently needs evaluation toolkits with evaluation partners to meet deployment-time requirements and better metrics for fair and comprehensive comparisons.

In this paper, we introduce **ZSC-Eval**, a comprehensive and convenient evaluation toolkit and benchmark, including the generation and selection of evaluation partners and measurement of ZSC capability with novel metrics. Inspired by reward hypothesis [52, 4], we assume that deployment-time partners' requirements can be represented as reward functions. Therefore, we use the widely adopted event-based reward functions [36, 11, 33] to indicate deployment-time partners' behavior preferences, which is practical for humans to designate evaluation partners [60]. To address the unbalanced distribution of generated partners and consequential unbalanced performance estimation, we propose Best Response Diversity (BR-Div), the population diversity [40] of partners' BRs, to select representative subsets as evaluation partners. For a comprehensive evaluation of generalization performance, we propose BR-Prox, which measures the performance similarity between ego agents and approximate BRs to the evaluation partners, illustrating the generalization gap and balancing evaluation partners with different cooperation capabilities.

We first verify the effectiveness of ZSC-Eval by demonstrating that the generated evaluation partners exhibit more diverse high-level behaviors than those in current evaluation methods. We then evaluate current ZSC algorithms using different evaluation methods and humans in the most popular coordination environment, Overcooked [5, 22] and show that ZSC-Eval can provide consistent results with human evaluation. We also provide benchmark results of current ZSC algorithms in Overcooked, in which we develop new testbeds. To verify the scalability of ZSC-Eval, we also provide benchmark results in Google Research Football (GRF) [20]. Through these experiments, we conclude guidelines for designing ZSC testbeds and further analyze the failure of current ZSC algorithms to generate enough diverse expert training partners.

In summary, our contributions are as follows: 1) To the best of our knowledge, we are the first to investigate the evaluation of ZSC capability and analyze the limitations of current evaluation methods; 2) We propose ZSC-Eval, a comprehensive and convenient evaluation toolkit and benchmark for ZSC algorithms, including partner candidates generation via behavior-preferring rewards, partners selection via BR-Div, and ZSC capability measurement via BR-Prox; 3) ZSC-Eval comprises human evaluation benchmark results from our human study platform, a part of ZSC-Eval, and comprehensive benchmark results with our generated evaluation partners, providing guidelines for designing ZSC testbeds and empirical analyses for current ZSC algorithms.

## 2 Related Work

**ZSC Problem and Methods.** ZSC algorithms aim to train an ego agent that can be deployed to coordinate with unseen partners without further training. Self-play (SP) [53, 59, 55] is a common way to train ego agents but learns conventions between players and generates agents that lack coordination with unseen partners [5]. Based on SP, representative algorithms involving game structure random-

Table 1: Comparison of evaluation partners used in recent works. *Cost* - Implementation efforts and financial outlay. *Extendability* - the degree to which they can be expanded in new scenarios. *Unseen* - their distributions are not similar to training partners. *Diverse* - their skills' style and level are diverse. *Deployment Requirements* - they can follow the distribution of deployment-time partners.

| Evaluation Partners | Reproducible | Cost | Extendability | Unseen | Diverse | Deployment Requirements |
|---|---|---|---|---|---|---|
| Human Players [5] | × | High | - | ✓ | ✓ | ✓ |
| Human Proxy Agents [5] | ✓ | Medium | Weak | ✓ | × | × |
| Trained Self-play Agents [51] | ✓ | Low | Strong | × | × | × |
| Trained Adaptable Agents [57] | ✓ | Low | Moderate | × | × | × |
| Rule-based Specialist [60] | ✓ | High | Weak | ✓ | ✓ | ✓ |
| Random Agents [51] | ✓ | Low | Strong | ✓ | × | × |
| **ZSC-Eval (Ours)** | **✓** | **Low** | **Strong** | **✓** | **✓** | **✓** |

ization [13] and diversity-based reward shaping [29] are derived to mitigate convention overfitting. Besides, Population-based training (PBT) algorithms [15], such as Population Play (PP) [5], train an ego agent that interacts within a population and encounters multiple partners during training. Fictitious co-play (FCP) [51] proposes a two-stage *Co-Play algorithm* involving self-play pre-training and ego agent training with the pre-trained population. Most co-play algorithms enhance the diversity of training population by population entropy-shaped reward [65], hidden-utility reward functions that model human behaviors [60], training incompatible agents [6], and contextual encoding for partner identification [28, 27]. Moreover, *Evolution algorithms* train the ego agent with evolving populations, updating the pool by promoting unique behaviors [30] and open-ended learning [57, 23, 24, 58].

**ZSC Evaluation and Analysis.** Researchers have analyzed ZSC in human-agent and agent-agent teams. McKee et al. [32] introduce the expected action variation metric for population diversity to assess agents' generalization. Knott et al. [17] argue that the average training or validation rewards do not reflect agent robustness. Some studies discuss the subjective evaluation of human-AI team performance but do not focus on ZSC capability [48, 31]. In contrast, we focus on evaluating the ZSC capability using diverse evaluation partners. Our ZSC-Eval aims to solve problems in generating evaluation partners and comprehensive and fair comparisons. We analyze the current evaluation methods and demonstrate the superiority of our ZSC-Eval in Table 1. To the best of our knowledge, ZSC-Eval is the first evaluation toolkit and benchmark for comprehensive ZSC capability evaluation.

# 3 Background

## 3.1 Decentralized Markov Decision Process

We formulate the ZSC problem in multi-agent scenarios as a decentralized Markov decision process (DEC-MDP) [3]. An $n$-agent DEC-MDP can be formalized as $< \mathcal{S}, \{\mathcal{A}^i\}_{i \in \mathcal{N}}, \rho, \mathcal{T}, r, \gamma >$, where $\mathcal{N} = \{1, \ldots, n\}$ is the set of agents, $\mathcal{S}$ is the state space, $\rho : \mathcal{S} \mapsto [0, 1]$ is the distribution of the initial state $s_0$. $\mathcal{A}^i$ is the action space of agent $i$, and $\mathcal{A} = \mathcal{A}^1 \times \cdots \times \mathcal{A}^n$ is the joint action space. $\mathcal{T} : \mathcal{S} \times \mathcal{A} \times \mathcal{S} \mapsto [0, 1]$ denotes the transition probability. $r : \mathcal{S} \times \mathcal{A} \mapsto \mathbb{R}$ is the reward function, and $\gamma \in [0, 1)$ is a reward discount factor. At time step $t$, each agent $i$ takes action $a_t^i$ from its policy $\pi^i(\cdot|s_t)$, simultaneously according to the state $s_t$, forming the joint action $\boldsymbol{a}_t = \{a_t^1, \ldots, a_t^n\}$ and the joint policy $\boldsymbol{\pi}(\cdot|s_t) = \pi^1 \times \ldots \times \pi^n$. We denote the expected discounted return as $\mathcal{J}(\boldsymbol{\pi}) = \mathbb{E}_{\tau \sim (\rho, \boldsymbol{\pi})} \left[ \sum_t \gamma^t r(s_t, \boldsymbol{a}_t) \right]$. Note that we concisely use $\mathcal{J}(\boldsymbol{\pi})$ under permutations of agents and $\mathcal{J}(\pi, \pi^{-i}) = \mathcal{J}(\pi, \ldots, \pi, \pi^{-i})$ where $\pi$ repeats $n - |\pi^{-i}|$ times, without loss of generality. For convenience, we denote the Best Response (BR) of policy $\pi^{-i}$ as $BR(\pi^{-i}) = \text{argmax}_{\pi'} \mathcal{J}(\pi', \pi^{-i})$. Let $\Pi_{\text{test}}$ be the set of potential unseen partners, named deployment-time partners in this paper, and $\pi^i$ be the ego agent's policy. The optimization objective of the ZSC problem can be represented as: $\max_\pi \mathbb{E}_{\mathbb{L} \sim \mathcal{U}(\mathcal{P}(\Pi_{\text{test}}))} \left[ \mathcal{J}(\pi, \{\pi^i\}_{i \in \mathbb{L}}) \right]$, where $\mathcal{P}(\mathbb{P}) = \{\mathbb{L} \in \mathbb{P}^m | 1 \leq m < n\}$ denotes the combinations of agents in $\mathbb{P}$ with different sizes, and we assume partners are sampled from a uniform distribution $\mathcal{U}$. As we focus on population-based ZSC algorithms, we further formalize the objective that considers the construction of the training population:

$$\max_{\Pi_{\text{train}}, \mathcal{O}} \mathbb{E}_{\mathbb{L} \sim \mathcal{U}(\mathcal{P}(\Pi_{\text{test}}))} \left[ \mathcal{J}\left( \mathcal{O}(\Pi_{\text{train}}), \{\pi^i\}_{i \in \mathbb{L}} \right) \right] ,$$

where $\Pi_{\text{train}}$ is the population constructed during training and $\mathcal{O}$ is an approximate oracle function that computes the common best response for partners in $\Pi_{\text{train}}$. For instance, the oracle function can be defined to maximize the objective with $\mathcal{U}(\Pi_{\text{train}})$, i.e., $\mathcal{O}(\Pi_{\text{train}}) = \arg\max_{\pi} \mathbb{E}_{\mathbb{L} \sim \mathcal{U}(\mathcal{P}(\Pi_{\text{train}}))} \left[ \mathcal{J}(\pi, \{\pi^i\}_{i \in \mathbb{L}}) \right]$.

## 3.2 Limitations of Current Evaluation Methods

In MARL, the ZSC problem focuses on zero-shot generalization to unseen cooperative partners, presenting an OOD generalization challenge due to the difference between training and deployment-time partners. To evaluate the ZSC capability accurately, evaluation partners must follow deployment-time partners' distributions, including human and other agents. Furthermore, the generalization performance of the ego agent must be measured in addition to task performance metrics like episode returns [16]. We discuss the gap between reasonable and current evaluation methods as follows.

**Are current evaluation partners convenient and following the distributions of deployment-time partners?** Diversity in evaluation partners is not just a desirable feature but a necessary condition for them to effectively cover the distribution of deployment-time partners. In our analysis, current evaluation partners in Table 1 (detailed in Appendix A) can be classified into two types: training-based and non-training-based. As the most commonly used training-based evaluation partners, human proxy agents trained by behavior cloning human data can not mimic real human behaviors [60]. Trained self-play partners frequently fail to diverge from the training population since they may not achieve distinct high-level behaviors [7, 45] though efforts have been made to generate diversity through low-level behavior optimization [30, 65]. Non-training-based partners like random agents do not provide diversity while maintaining high performance. The lack of diversity among these partners makes it difficult to match the distribution of deployment-time partners. Besides, some evaluation partners suffer from reproducibility problems, high implementation costs, and extendability problems, and they are similar to training partners, as summarized in Table 1.

**Can current evaluation methods and metrics demonstrate the ZSC capability?** At present, evaluation methods in ZSC can be broadly classified into two categories: evaluations with fixed partners and cross-play evaluations. Cross-play evaluations, i.e., using trained adaptive agents from ZSC algorithms as evaluation partners and cross-playing the agents to compare the performance mutually, risk unfair comparisons due to overlaps between evaluation and training partners. Moreover, eliminating overlapping partners might compromise the control conditions of experiments. As for evaluation metrics, both approaches use mean episode returns to evaluate the ZSC capability. However, the current metric needs revision to measure the overall generalization performance of ZSC. It fails to capture crucial aspects such as the generalization gap [16] and ignores different cooperation capabilities among evaluation partners, highlighting the need for more comprehensive evaluation metrics. The potential for unfair comparisons and limitations of current evaluation metrics significantly undermine their effectiveness in assessing ZSC capabilities.

In summary, there is an urgent need in the ZSC community to develop a comprehensive evaluation toolkit and benchmark to assess ZSC capability more accurately and drive progress in ZSC.

## 4 ZSC-Eval

As shown in Figure 1, ZSC-Eval includes evaluation partners generation and selection, and ZSC capability measurement.

### 4.1 Generation of Behavior-preferring Agents as Candidates

Based on the reward hypothesis that goals and purposes can be well thought of as maximizing the expected cumulative sum of the received reward [52, 4], we assume that requirements for deployment-time partners can be represented as reward functions $\mathcal{R}^{\text{Deploy}} = \{r_1, \ldots, r_P\}$, where $P$ is the number of partners. Consequently, deployment-time partners can be approximated by optimizing policies to maximize these reward functions. Specifically, the distribution of deployment-time partners is tailored to various applications, such as care robots [41, 37] and team sports [12]. Event-based rewards are widely adopted as a standard reward design method and a practical design principle in these applications [36, 11, 33, 62, 2, 47]. Therefore, we use event-based rewards, which we named behavior-preferring rewards, to approximate $\mathcal{R}^{\text{Deploy}}$. Behavior-preferring rewards allow conveniently

---

**Algorithm 1: Evaluation Partners Generation and Selection**

**Input:** Reward Space $\mathcal{R}^{\text{BP}}$, Number of Candidates $N$, Number of Evaluation Partners $M$.

**Output:** Evaluation Partners $\{\pi_{\boldsymbol{w}^i}\}_{i\in\mathbb{S}}$ and Best Responses $\{\widehat{BR}(\{\pi_{w^i}\}_{i\in\mathbb{L}})\}_{\mathbb{L}\in\mathcal{P}(\mathbb{S})}$.

1 **for** $i = 1, \ldots, N$ **do**
2     Sample a behavior-preferring reward function $r_{\boldsymbol{w}^i}$ from $\mathcal{R}^{\text{BP}}$.
3     Obtain $(\pi_{\boldsymbol{w}^i}, \widehat{BR}(\pi_{\boldsymbol{w}^i}))$ by performing PPO independently to solve the Markov Game.
4     Evaluate $(\pi_{\boldsymbol{w}^i}, \widehat{BR}(\pi_{\boldsymbol{w}^i}))$ and embed $\widehat{BR}(\pi_{\boldsymbol{w}^i})$'s high-level behavior features as $\boldsymbol{\theta}_{\boldsymbol{w}^i}$.
5 Compute the similarity matrix of $\{\widehat{BR}(\pi_{\boldsymbol{w}^i})\}_{i=1}^N$ as $\boldsymbol{K}$.
6 Sample a subset $\mathbb{S}$ of size $M$ by Determinantal Point Proces sampling with $\boldsymbol{K}$.
7 Select checkpoints as $\mathbb{C}$ and update $\mathbb{S} = \mathbb{S} \cup \mathbb{C}$.
8 Train approximate BRs $\{\widehat{BR}(\{\pi_{w^i}\}_{i\in\mathbb{L}})\}_{\mathbb{L}\in\mathcal{P}(\mathbb{S})}$.

---

designating the coverage of evaluation partners to include common and edge cases [17] and entail high reproducibility, low implementation cost, and strong extendability, as summarized in Table 1.

Specifically, we use a linear function combination to approximate the reward space $\mathcal{R}^{\text{Deploy}}$, as in Ng et al. [36]. The approximate reward space is defined as $\mathcal{R}^{\text{BP}} = \{r_{\boldsymbol{w}} | r_{\boldsymbol{w}}(s_t, \boldsymbol{a}_t) = r + \phi(s_t, \boldsymbol{a}_t)^T\boldsymbol{w}, \boldsymbol{w} \in \mathbb{R}^m, \|\boldsymbol{w}\|_\infty \leq B_{\max}, \sum_i \mathbb{1}(\boldsymbol{w}_i \neq 0) \leq C_{\max}\}$, where $\boldsymbol{w}$ is an $m$-dimensional weight vector and $r_{\boldsymbol{w}}$ is the reward function that encourages behaviors indicated by $\boldsymbol{w}$. $\phi : \mathcal{S} \times \mathcal{A} \mapsto \mathbb{R}^m$ embeds event-based features, e.g., $\phi(s_t, \boldsymbol{a}_t)_j$ indicates whether the $j$-th event has occurred. $B_{\max}$ limits the norm of $\boldsymbol{w}$, while $C_{\max}$ limits the number of events, eliminating unusual behaviors. The original game reward $r$ is added to prevent behavior-preferring agents from sabotaging. Under these constraints, $\mathcal{R}^{\text{BP}}$ promotes diverse behaviors and still encourages cooperative task completion.

We train behavior-preferring agents and their best responses using behavior-preferring reward functions. Given a specific reward function $r_{\boldsymbol{w}}$, one player receives this reward while the others continue receiving the original game reward $r$. The procedure of optimizing the players' objectives can be formulated as finding a Nash Equilibrium (NE) [38] in a Stochastic Game [54]. We can approximate an NE by agents independently performing the Proximal Policy Optimization (PPO) algorithm [46] since $r_{\boldsymbol{w}} \in \mathcal{R}^{\text{BP}}$ still guides the behavior-preferring agent to cooperate for solving the given task [8]. After approximating an NE, we obtain $\pi_{\boldsymbol{w}}$ that learns the behaviors preferred by $\boldsymbol{w}$ and $\widehat{BR}(\pi_{\boldsymbol{w}})$, the approximate BR of $\pi_{\boldsymbol{w}}$. Line 1 to Line 3 in Algorithm 1 summarize that ZSC-Eval constructs evaluation partner candidates which cover a set of diverse behaviors by sampling reward functions from $\mathcal{R}^{\text{BP}}$ and approximating NEs.

### 4.2 Selection of Evaluation Partners by Best Response Diversity

The generated candidates may be unbalanced distributed in cooperative conventions and behaviors, which we further discuss in Section 5.1. To avoid the unbalanced evaluation of ZSC agents that coordinate well with those behaviors with high proportions in candidates, we need to select a representative subset of candidates as evaluation partners.

Typically, the most representative population subsets can be obtained by maximizing the *population diversity* [40]. We first define the *population diversity* of a population $\{\pi_i\}_{i=1}^M$

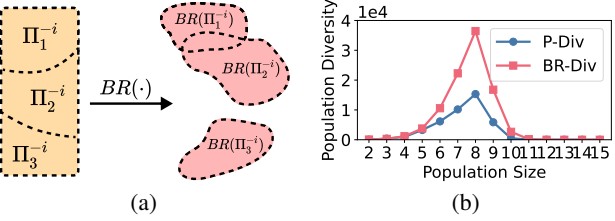

Figure 2: (a) Different partners may respond to similar BRs. (b) Population diversity of BRs to partner subsets selected by two methods with different sizes. A higher vertical axis value at the same subset size indicates more diverse BRs in the subset.

as the determinant of the population's similarity matrix: $\text{PD}(\{\pi_i\}_{i=1}^M) := \det(\boldsymbol{K})$, where $\boldsymbol{K}_{ij} = \boldsymbol{\theta}_i \cdot \boldsymbol{\theta}_j$ is the similarity matrix of the population, and $\boldsymbol{\theta}_i$ is the behavior feature of policy $\pi_i$.[3] One can intuitively repeat sampling subsets from candidates and

---

[3]For simplicity, we count the occurrences of events during episodes as the policy behavior.

select the subset with the maximum population diversity as the evaluation partners, which can be formatted as maximizing the *Partner Diversity* (P-Div), where $\text{P-Div}(\{\pi_i\}_{i=1}^M) = \text{PD}(\{\pi_i\}_{i=1}^M)$.

However, based on the fact that an ego agent with strong ZSC capability should emulate any policy in the set of BRs to evaluation partners [30], the evaluation method should expose the ego agent to evaluation partners with diverse BRs. Different partners selected by P-Div may respond to similar BRs [45, 22], as illustrated in Figure 2(a). Therefore, maximizing P-Div may not necessarily produce partners that require diverse skills to coordinate with [42, 43]. To further verify, we define **Best Response Diversity** (BR-Div) as $\text{BR-Div}(\{\pi_i\}_{i=1}^M) := \text{PD}(\{\widehat{BR}(\pi_i)\}_{i=1}^M)$, which is the population diversity of approximate BRs to selecte candidates. As in Figure 2(b), selections from a pool of evaluation partner candidates based on BR-Div reach a higher population diversity of BRs than those based on P-Div, meaning that maximizing BR-Div is more effective in constructing evaluation partners with diverse BRs. We include details of Figure 2(b) and demonstrate that evaluation partners selected by BR-Div exhibit more diverse behaviors than those selected by P-Div in Appendix D.

Therefore, we select evaluation partners through maximizing BR-Div, as summarized in Line 4 to Line 6 in Algorithm 1. In detail, we count occurrences of pre-defined events of $\widehat{BR}(\pi_{\boldsymbol{w}^i})$ alongside episodes as the high-level behavior feature $\boldsymbol{\theta}_{\boldsymbol{w}^i} = \mathbb{E}_{\pi_{\boldsymbol{w}^i}, \widehat{BR}(\pi_{\boldsymbol{w}^i})}[\sum_{t=1}^T \phi(s_t, \boldsymbol{a}_t)] \in \mathbb{R}^m$ of $\widehat{BR}(\pi_{\boldsymbol{w}^i})$ for calculating BR-Div. Then we compute the similarity matrix as $\boldsymbol{K}$ where $\boldsymbol{K}_{ij} = \boldsymbol{\theta}_{\boldsymbol{w}^i} \cdot \boldsymbol{\theta}_{\boldsymbol{w}^j}$. Since BR-Div is defined as a determinant function, we apply the Determinantal Point Process (DPP) [19] to search for the candidate subset of size $M$ with the maximum determinant. DPP samples proportionally to determinants of candidate subsets: $P(\{\pi_{\boldsymbol{w}^i}\}_{i \in \mathbb{P}}) \propto \text{BR-Div}(\{\pi_{\boldsymbol{w}^i}\}_{i \in \mathbb{P}}) = \det(\boldsymbol{K}_{\mathbb{P}})$, where $\mathbb{P} \subset \{1, \dots, N\}$ is the subset's indices and $\boldsymbol{K}_{\mathbb{P}}$ denotes the submatrix of $\boldsymbol{K}$ obtained by restricting rows and columns indexed in $\mathbb{P}$. Because candidate subsets are usually inexhaustible, we repeat DPP sampling to search for the representative candidate subset and denote the selected subset as $\mathbb{S} = \text{argmax}_{\mathbb{P}} \text{BR-Div}(\{\pi_{\boldsymbol{w}^i}\}_{i \in \mathbb{P}})$ and $|\mathbb{P}| = M$. Furthermore, as shown in Line 7 of Algorithm 1, we collect the earlier checkpoints of selected candidates to enhance the diversity of skill levels, which satisfies $\mathcal{J}(\widehat{BR}(\dot{\pi}_{\boldsymbol{w}^i}), \dot{\pi}_{\boldsymbol{w}^i}) \approx \mathcal{J}(\widehat{BR}(\pi_{\boldsymbol{w}^i}), \pi_{\boldsymbol{w}^i})/2, \ i \in \mathbb{S}$.

### 4.3 Measurement of ZSC Capability by Best Response Proximity

Previous evaluation methods measure ZSC capability by mean episode returns, but there are two limitations to using mean episode returns: 1) Using mean episode returns does not provide a standard for presenting how well the learned cooperation ability is generalized. For example, when evaluating agents' generalization ability, it is recommended to show their generalization gap in auxiliary, i.e., the gap between training performance and testing performance [16]. 2) Mean episode returns do not consider the unbalanced cooperation capabilities among evaluation partners, and results with different evaluation partners should not be weighted equally. To tackle these limitations, we introduce the **Best Response Proximity** (BR-Prox) metric. Formally, we define:

$$\text{BR-Prox}\left(\pi, \{\pi_{\boldsymbol{w}^i}\}_{i \in \mathbb{P}}\right) := \underset{\mathbb{L} \in \mathcal{P}(\mathbb{P})}{\text{Aggr}} \left( \mathcal{J}(\pi, \{\pi_{\boldsymbol{w}^i}\}_{i \in \mathbb{L}}) \ / \ \mathcal{J}(\widehat{BR}(\{\pi_{\boldsymbol{w}^i}\}_{i \in \mathbb{L}}), \{\pi_{\boldsymbol{w}^i}\}_{i \in \mathbb{L}}) \right),$$

where $\text{Aggr}$ means the aggregator across evaluation partners, such as the most common 'mean' and 'median' aggregators. We adopt the inter-quartile mean to aggregate the data [1], focusing on the middle 50% for statistical reliability. BR-Prox evaluates performance similarity between the ego agent and approximate BRs, presenting the generalization gap and balancing results with evaluation partners based on their cooperation capability. Since a single score cannot fully capture the performance variance across evaluation partners [17], we recommend reporting the results with 95% confidence intervals [35] and inter-quartile ranges, e.g., the middle 50% of disaggregated scores.

## 5 Experiments

In this section, we conduct a series of experiments in popular coordination environments Overcooked [5, 22] and Google Research Football (GRF) [20]. We first verify ZSC-Eval's effectiveness both in generating diverse evaluation partners and evaluating ZSC capability, compared with current evaluation methods, including human evaluation. We then benchmark current ZSC algorithms using ZSC-Eval and show novel empirical findings about how ZSC-Eval helps evaluate ZSC algorithms.

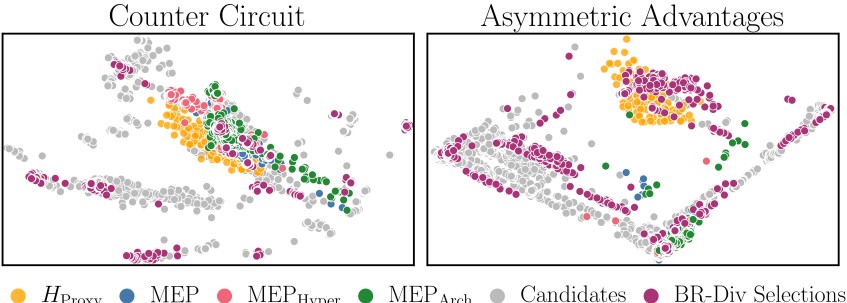

Figure 3: Visualization of high-level behaviors of human proxy agents, different self-play populations, our evaluation partner candidates, and evaluation partners in Overcooked layouts.

**Environments.** We conduct experiments in two environments. We retain four commonly used layouts in Overcooked, including Asymm. Adv., Coord. Ring, Forced Coord., and Counter Circ.. We leverage two new layouts, Bothway Coord. and Blocked Corr. and create three more new layouts with the multi-recipe mechanism to increase the necessity and difficulty of cooperation. Then, we choose the '3 vs 1 with Keeper' scenario in GRF as a ZSC testbed, letting the ego agent be a team member and collaborate with the other three players. Environment details can be found in Appendix B.

**Experiment Setup.** We implement six strong methods, including FCP [51], MEP [65], TrajeDi [30], HSP [60], COLE [23] and E3T [58], and additionally add self-play (SP) [5] as a baseline. We also evaluate these ZSC algorithms with humans in Overcooked. More experiment setup details and full results are in Appendix C.

## 5.1 Effectiveness of ZSC-Eval

**ZSC-Eval's Evaluation Partners Exhibit the Most Diverse Behaviors.** We demonstrate the population diversity of our generated evaluation partners and evaluation partners used in current evaluation methods in Figure 3. Our generated evaluation partners exhibit the most diverse behaviors. More results shown in Appendices D.2 and D.3 further illustrate that our generated evaluation partners exhibit more diversity in high-level behaviors and episode return distributions obtained with ZSC agents. The diversity of evaluation partners means that ZSC-Eval has a strong ability to approximate deployment-time partners.

**Highly Similarity between Evaluations by ZSC-Eval and Human.** To demonstrate the effectiveness of ZSC-Eval for evaluating ZSC algorithms, we compare human evaluation results with evaluation results using different evaluation partners. The results shown in Table 2 verify that ZSC-Eval's results are the closest to human evaluations and that the Spearman's rank correlation coefficient ($r_s$) [49] between ZSC-Eval and human evaluation reaches the highest, meaning that ZSC-Eval effectively obtains evaluation results similar to those with humans. We also collect human subjective rankings and compare them with objective score rankings. The human subjective perceptions are generally consistent with the objective episode returns. Detailed results are provided in the Appendix D.7.

Table 2: Ranks of ZSC algorithms under different evaluation partners in various Overcooked layouts. $r_s$ measures the correlation between ranks under human evaluation and ranks under others.

| Layouts | Eval Partners | ZSC Algorithms | | | | | $r_s$ |
|---|---|---|---|---|---|---|---|
| | | HSP | MEP | FCP | COLE | SP | |
| Coord. Ring | Human | 3 | 1 | 2 | 4 | 5 | - |
| | ZSC-Eval (Ours) | 2 | 1 | 3 | 4 | 5 | **0.90** |
| | Human Proxy | 2 | 1 | 3 | 4 | 5 | **0.90** |
| | Trained SP Agents | 4 | 1 | 3 | 2 | 5 | 0.70 |
| Counter Circ. | Human | 1 | 3 | 2 | 4 | 5 | - |
| | ZSC-Eval (Ours) | 1 | 3 | 2 | 4 | 5 | **1.00** |
| | Human Proxy | 3 | 1 | 2 | 4 | 5 | 0.60 |
| | Trained SP Agents | 4 | 3 | 2 | 1 | 5 | 0.10 |

## 5.2 Benchmark Results and Empirical Findings in Overcooked

We present abundant benchmark results with 9 Overcooked layouts in Figures 4 to 6, in which we implement each population-based algorithm with three different population sizes. We observe that co-play algorithms outperform other algorithms in most layouts, and population-based algorithms

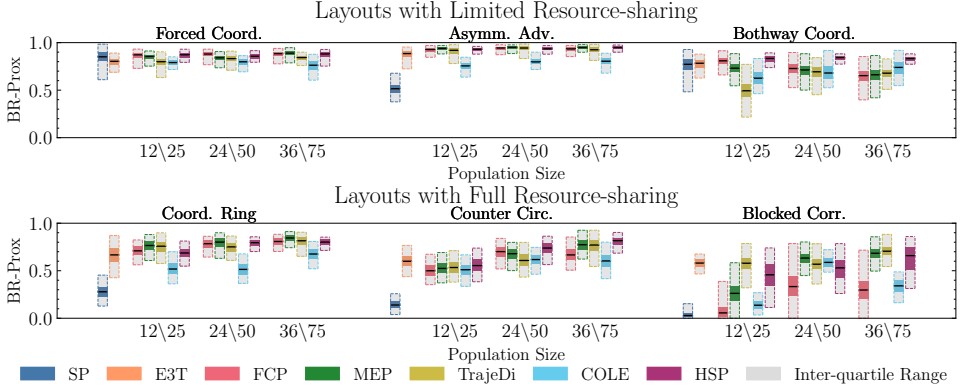

Figure 4: BR-Prox performance with 95% confidence intervals of ZSC algorithms with different population sizes in Overcooked. '12\25', '24\50' and '36\75' mean that co-play methods (FCP, MEP, TrajeDi and HSP) are trained with populations of 12, 24 and 36 and that the evolution method (COLE) is trained with populations of 25, 50 and 75. Note that SP and E3T are not population-based.

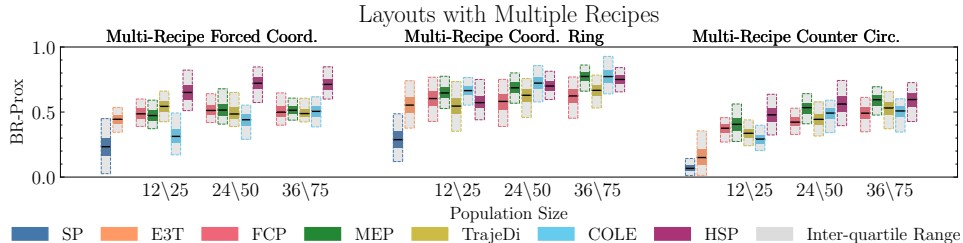

Figure 5: BR-Prox performance of ZSC algorithms in Overcooked with multiple recipes.

generally perform better as the population size increases. Full results can be found in Appendix D. We summarize two empirical findings below.

**Guidelines for Increasing Complexity in Designing ZSC testbeds.** Results in Figure 4 indicate the commonly used layouts, Forced Coord. and Asymm. Adv. fail to differentiate algorithms' performance. We have also noticed that SP performs well in these layouts, indicating that it can easily learn most of the skills for interacting with unseen partners. These results suggest that some layouts' simplistic design limits the showcase of agents' ZSC capabilities due to insufficient required cooperative strategies. We improve layouts by increasing their complexity, including the complexity of agent coordination (*coordination complexity*) and overall team tasks (*task complexity*).

*Coordination Complexity.* We observe the connection between whether layouts differentiate algorithms' performance and degrees of resource-sharing in four old layouts and then classify the layouts as 'Limited Resource-sharing' and 'Full Resource-sharing' in Figure 4. As a case of coordination complexity, the resource-sharing mechanism increases the need for cooperative strategies and helps measure ZSC capability, demonstrating the importance of increasing coordination complexity. To investigate further, we increase the coordination complexity by letting the layouts require more frequent interaction among agents. The Bothway Coord. and Blocked Coor. layouts we leverage include passing cooking ingredients bidirectionally and scheduling spare counters and a corridor. In these new layouts, ZSC algorithms exhibit a more significant performance difference, demonstrating the effectiveness of increasing coordination complexity.

*Task Complexity.* We leverage the multi-recipe mechanism in three old layouts to increase the task complexity and then present benchmark results in these new layouts. As shown in Figure 5, the performance difference between ZSC algorithms has significantly increased after using the multi-recipe mechanism, indicating the effectiveness of increasing task complexity. While the increased population size could improve policy diversity within the population [32], the performance improvement as the population size grows is only apparent in experiments where we leverage the multi-recipe mechanism. Such a phenomenon indicates that the increased task complexity enables layouts to demonstrate the effect of varying population sizes since more cooperative strategies are

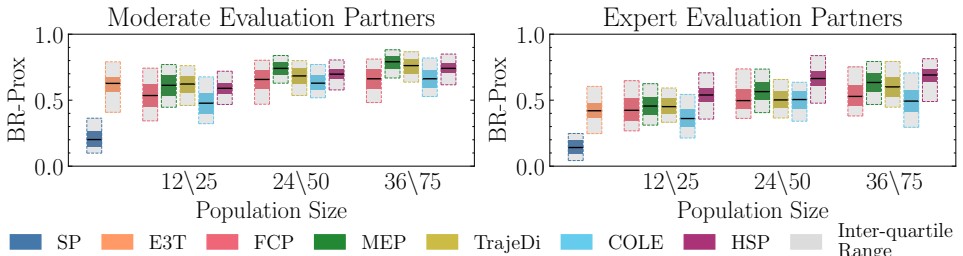

Figure 6: We compare the aggregated BR-Prox performance obtained with evaluation partners at different skill levels.

required, which is desired for ZSC evaluation. Therefore, when developing ZSC testbeds, we suggest prioritizing task complexity.

**Performance Degradation with Expert Evaluation Partners.** ZSC-Eval highlights the performance variation of ZSC algorithms with evaluation partners of varying skill levels. We investigate how ZSC algorithms perform when faced with unseen partners at different skill levels by considering the evaluation partners with self-play performance less than the median, i.e., $\{\pi^P | \pi^P \in \mathbb{S}, \mathcal{J}(\pi^P) \leq \text{Median}_{\pi^{P'} \in \mathbb{S}} \mathcal{J}(\pi^{P'})\}$, as moderate evaluation partners, and the left ones as expert evaluation partners. Owing to BR-Prox that measures generalization capability rather than episode returns, ZSC-Eval reveals that current ZSC algorithms perform worse with expert evaluation partners, as shown in Figure 6. Furthermore, increasing population size has a lower impact on performance when dealing with expert evaluation partners than moderate evaluation partners. Such results may result from current ZSC algorithms failing to generate enough diverse expert agents even with increasing population sizes, which can be diagnosed using our proposed BR-Div, as elaborated in Appendix D.4.

### 5.3 Evaluating Zero-shot Coordination Capability in Google Research Football

We further evaluate current ZSC algorithms by ZSC-Eval in the GRF academy '3 vs 1 with Keeper' scenario, a complex cooperative environment with a large state-action space and strong built-in bots as opponents, to investigate ZSC-Eval's scalability. Table 3 shows the performance of each method playing the three-player football game with our evaluation partners, in which the ZSC algorithms' ranks are similar to those in Overcooked. In Appendix D.2, we further illustrate diverse high-level behaviors of ZSC-Eval generated evaluation partners in GRF. These results verify that ZSC-Eval can conveniently scale to more complex scenarios with more than two players.

Table 3: BR-Prox performance with 95% confidence intervals of ZSC algorithms in GRF.

| Method | BR-Prox | (95% CI) |
|--------|---------|----------|
| SP | 0.20 | (0.14, 0.24) |
| E3T | 0.66 | (0.59, 0.74) |
| FCP | 0.78 | (0.72, 0.85) |
| MEP | 0.78 | (0.71, 0.85) |
| TrajeDi | **0.81** | **(0.75, 0.89)** |
| COLE | 0.75 | (0.69, 0.84) |
| HSP | **0.80** | **(0.72, 0.88)** |

## 6 Conclusion

In this paper, we first analyze problems of current ZSC evaluation methods, particularly mismatched distributions between evaluation and deployment-time partners and inadequacy metrics for measuring ZSC capability. We present ZSC-Eval, a toolkit and benchmark for evaluating ZSC algorithms, which includes: 1) evaluation partner candidates generation via behavior-preferring rewards, 2) evaluation partners selection via BR-Div, and 3) ZSC capabilities measurement via BR-Prox. ZSC-Eval includes Overcooked and GRF as testbeds and implements commonly used ZSC algorithms. Although ZSC-Eval has limitations in fully representing deployment-time partners, we demonstrate its effectiveness by verifying the diversity of generated evaluation partners and the consistency between ZSC-Eval 's evaluation results and human evaluation results. Another limitation mainly lies in the fact that the design of event-based rewards needs careful handcraft and that event-based rewards may not fully represent deployment-time partners. The events represent various situations during the deployment time, which requires the designer to have a comprehensive understanding of the environments and tasks, and is hard to exhaust. The limitation results from the challenge of reward design, which is an inherent challenge in reinforcement learning [26]. To alleviate these limitations, a promising

further direction is to leverage some automatic reward design techniques, e.g., leveraging the large language models for reward generation [21]. We create new ZSC testbeds, propose guidelines for designing ZSC testbeds, and provide detailed analyses about the failure of current ZSC algorithms in coordinating with expert evaluation partners. We believe that ZSC-Eval could be a convenient scaffold for developing future ZSC algorithms.

## Acknowledgement

This work is partially supported by National Key R&D Program of China (2022ZD0114804), Shanghai Municipal Science and Technology Major Project (2021SHZDZX0102) and National Natural Science Foundation of China (62322603, 62076161, 62106141). Xihuai Wang and Jingxiao Chen are both supported by the Wen-Tsun Wu AI Honorary Doctoral Scholarship from AI Institute, Shanghai Jiao Tong University. The authors thank Yang Li for his help in the paper writing. The authors extend heartfelt thanks to the participants from Shanghai Jiao Tong University.

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

# A  Comparisons among Evaluation Methods

We categorize the partner agents involved in the evaluation of ZSC algorithms in Table 4, and offer a comprehensive analysis of the limitations associated with these partners.

Table 4: Evaluation partners used in recent works under the Overcooked environment.

| Evaluation Methods | Utilized by |
|---|---|
| Human Players | Overcook-AI [5]; MEP [65]; FCP [51]; HSP [60]; PECAN [28]; HiPT [27]; COLE [24]; E3T [58] |
| Human Proxy Agents | Overcook-AI [5]; MAZE [57]; FCP [51]; MEP [65]; HSP [60]; PECAN [28]; COLE [23]; HiPT [27]; E3T [58] |
| Trained Self-play Agents | FCP [51], MAZE [57], LIPO [6], HiPT [27] |
| Trained Adaptable Agents | MAZE [57], COLE [23], E3T [58] |
| Rule-based Specialist | HSP [60], LIPO [6] |
| Random Agents | FCP [51], MAZE [57] |

**Human Players.** The human player in ZSC problem is a type of 'perfect' evaluation partners, because humans are strictly qualify as unseen partners and represent the deployment-time requirements. Many works present a human evaluation as a main contribution [5, 51, 65, 60, 24, 28, 27]. The most challenge is that during the training process, a long-cycle human evaluation is not repeatable and cannot be replicated in large numbers to advance algorithm iterations. And the cost of human evaluation is also cannot be ignored. We need a more efficient evaluation method as a supplement for human evaluation.

**Human Proxy Agents.** The most widely used evaluation partners is human proxy agents proposed by [5]. The human proxy agents were trained by imitation learning method with a human datasets, which aims to represent human behaviors and human diversity [5]. And [51] used a similar way to construct a human proxy agents pool for evaluation. The availability and cost of human data are constrained, which are similar to the human evaluation. Yu et al. [60] highlighted that these human proxy agents in overcooked environment do not account for human behaviors, which shows that using human proxy agents does not represent the diversity from humans and does not validate the ZSC capabilities of the algorithm and fully compare various methods.

**Trained Self-play Agents.** The trained self-play agents are also a widely used partners for ZSC capability evaluation [51, 57, 6, 27]. However, through our experiment that comparing the similarities between two SP agents pools using different seeds (refer to Figure 14), we find that the diverse SP pool for evaluation which constructed using same algorithm in pre-training stage is similar to the pre-trained population in co-play methods. This flaw makes it difficult for these trained self-play agents to meet unseen requirements.

**Trained Adapted Agents.** The evaluation via cross-play with other trained adapted agents is inevitable to have a part of evaluation that is tested on algorithm's own training set (even including their own ego) [6, 57, 23]. The own training set even including own ego may leads to a higher performance and they are not unseen partners. Therefore, the performance of cross-play does not completely reflect the capabilities of ZSC, and may cause the performance to be falsely improved. And if excluding the performance from own training population to avoid the seen partners, the cross-play evaluation leads to a potential unfairness.

**Rule-based Specialist.** As a controllable method, using rule-based agents to evaluate the ZSC capability is also been used by some studies [60, 6]. The most problem is that compared to other methods, ruled-based agents evaluation is not extendable. Manually building expert rules is difficult to implement in complex environments and may not meet diversity requirements.

**Random Agents.** Another choice is using random initial agents as evaluation partners [51, 57]. However, the diversity of the random initialization cannot be ensure. And evaluate agent diversity is not only presenting in low level behaviors but also need a high level performance [7]. Random initialization lacks of a high level performance to ensure the evaluation pool is diverse enough.

The lack of diversity in random evaluation partners makes it difficult to represent deployment-time requirements, failing to comprehensively demonstrate ZSC capabilities.

**ZSC-Eval.** We remark that designing rewards to encourage desired behaviors requires much less implementation efforts and exhibits stronger extendability than implementing rule-based specialists since the former does not implement policies directly, although both of them requires human efforts.

# B    Experiment Environment

## B.1    Overcooked Environment

We re-evaluate in the Overcooked environment [5]. Overcooked is a simulation environment for reinforcement learning derived from the Overcooked! video game and popular for coordination problems [22]. The Overcooked environment features a two-player collaborative game structure with shared rewards, where each player assumes the role of a chef in a kitchen, working together to prepare and serve soup for a team reward. We retained 4 layouts including Asymmetric Advantages (Asymm. Adv.), Coordination Ring (Coord. Ring), Forced Coordination (Forced Coord.), and Counter Circuit (Counter Circ.) and added 3 new layouts: Bothway Coordination (Bothway Coord.), Blocked Corridor (Blocked Corr.) and Asymmetric Coordination (Asymm. Coord.). The figure of these layouts can be found in Figure 7. We further implement the multi-recipe mechanism in Forced Coordination (Forced Coord.), Coordination Ring (Coord. Ring) and Counter Circuit (Counter Circ.) layouts. As shown in Figure 8, the multi-recipe mechanism has onion (O) and tomatoes (T) as ingredients, which expands the range of recipes from just onion soup (3O) to five types of soups, including mix soup (1O1T), less onion soup (2O), tomato-onion soup (2T1O), onion-tomato soup (2O1T), and onion soup (3O).

Belows are the details and main challenges for each layout:

**Forced Coordination.** The Forced Coordination environment is designed to necessitate cooperation between the two players, as they are situated in separate, non-overlapping sections of the kitchen. Furthermore, the available equipment is distributed between these two areas, with ingredients and plates located in the left section and pots and the serving area in the right section. Consequently, the players must work together and coordinate their actions to complete a recipe and earn rewards successfully.

**Counter Circuit.** The Counter Circuit layout features a ring-shaped kitchen with a central, elongated table and a circular path between the table and the operational area. In this configuration, pots, onions, plates, and serving spots are positioned in four distinct directions within the operational area. Players may find themselves obstructed by narrow aisles, prompting the need for coordination to maximize rewards. One example of an advanced technique players can learn is to place onions in the middle area for quick and efficient passing, thereby enhancing overall performance.

**Asymmetric Advantages.** In the Asymmetric Advantages layout, players are divided into two separate areas, but each player can independently complete the cooking process in their respective areas without cooperation. However, the asymmetrical arrangement of the left and right sides encourages collaboration to achieve higher rewards. Specifically, two pots are placed in the central area, accessible to both players. The areas for serving and ingredients, however, are completely distinct. The serving pot is placed near the middle on the left side and far from the middle on the right side, with the ingredients area arranged oppositely. Players can minimize their walking time and improve overall efficiency by learning how to collaborate effectively.

**Coordination Ring.** The Coordination Ring layout is another ring-shaped kitchen, similar to the Counter Circuit. However, this layout is considerably smaller than Counter Circuit, with a close arrangement that makes it easier for players to complete soups. The ingredients, serving area, and plates are all in the bottom left corner, while the two pots are in the top right. As a result, this layout allows more easily achieving high rewards.

**Bothway Coordination.** Compared to the Forced Coordination, Bothway Coordination enables both left and right agents to have access to onions and pots, giving them more policy space and cooperation forms, which decreases the long waiting time in Forced Coordination and enriches their policy diversity. Meanwhile, the plates and the serving spot are still placed to one side, thus the two players still need to cooperate to finish an order.

**Blocked Corridor.** In the Block Corridor layout, the most challenging part is the corridor which is the only connection between the left and right parts with the small throughput of one person in the middle. Both onions and plates are placed at the upper edge of U-Shape corridor, while pot and serving spot are placed at two bottom corner. If there is no cooperation at all, the onion need to be carried from upper left to lower right while the teammate needs to stay at the spare place at right side to avoid conflict. If we want to implement cooperation, there are a lot of options of spare counter, which provides many alternatives for how to cooperate. The agent needs to show its diversity and be able to response well to all possible behaviors of the player. Additionally, conflicting positions within small corridors is a challenge that needs to be addressed. Definitely, it is the most challenging layout of our setup.

**Asymmetric Coordination.** Modified from Asymmetric Advantage, this layout expands the map and changes the plates to be asymmetric. The first change expand the trajectory space. The second change is that we make the right player have a shorter distance to pick a plate while the left player have a shorter to serve the soup, yielding a new cooperation form where right player pass the plate to left through the center counter.

**Forced Coordination with Multi-recipe.** Multi-recipe Forced Coordination is modified from Forced Coordination with the addition of tomatoes (another ingredient) on a shared counter between two players. In this and the following layouts, we've added multiple recipes with different rewind times and rewards. Two players need to complete a variety of orders within the rewind time, and cooperation is required in the process.

**Coordination Ring with Multi-recipe.** Similar with Forced Coordination with Multi-recipe, Coordination Ring with Multi-recipe adds tomatoes at the bottom-left locations near the serving area, which is a more complex version of Coordination Ring and increases the importance of cooperation.

**Counter Circuit with Multi-recipe.** Counter Circuit with Multi-recipe has the same layout as the Counter Circuit mentioned above. But it adds the multi-recipe settings as well. Due to the difficulty of the layout, we chose to keep onions as the only ingredient and try recipes with different amounts of onions to enrich the environment.

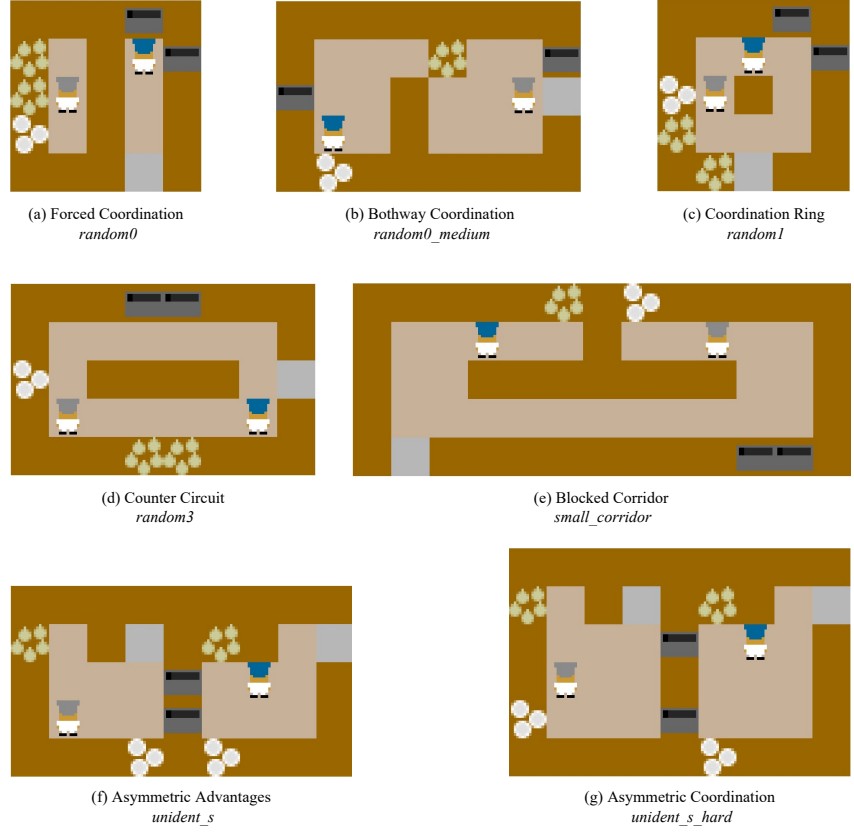

Figure 7: Used layouts in Overcooked.

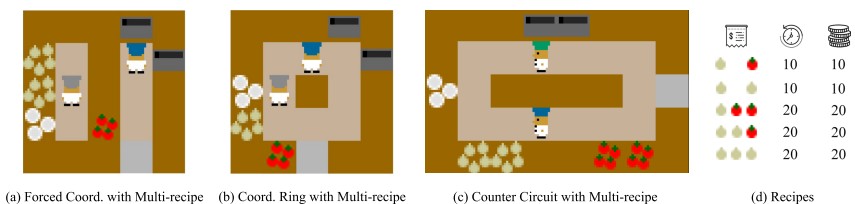

Figure 8: Multi-recipe Mechanism in Overcooked.

### B.2  Google Research Football

Google Research Football (GRF) [20] is a simulation environment for reinforcement learning based on the popular football video game. The GRF environment offers a multi-agent game setting with competitive or cooperative rewards, where each agent controls a football player in a realistic 3D stadium, trying to score goals and prevent the opponent from scoring. It features a continuous viewing space, comes in a variety of candidate formats including raw pixels, super mini maps, and floating vectors, and offers 19 discrete actions for each individual player. We choose the Football Academy 3 vs. 1 with Keeper scenario and implement it as a ZSC challenge.

**Google Research Football Academy 3 vs. 1 with Keeper scenario.** In this environment, three of our players try to score from the edge of the box, one on each side, and the other at the center. Initially, the player at the center has the ball, and is facing the defender. The defender side has a goal keeper. The offensive players need to cooperate by passing, dribbling or moving to score a goal.

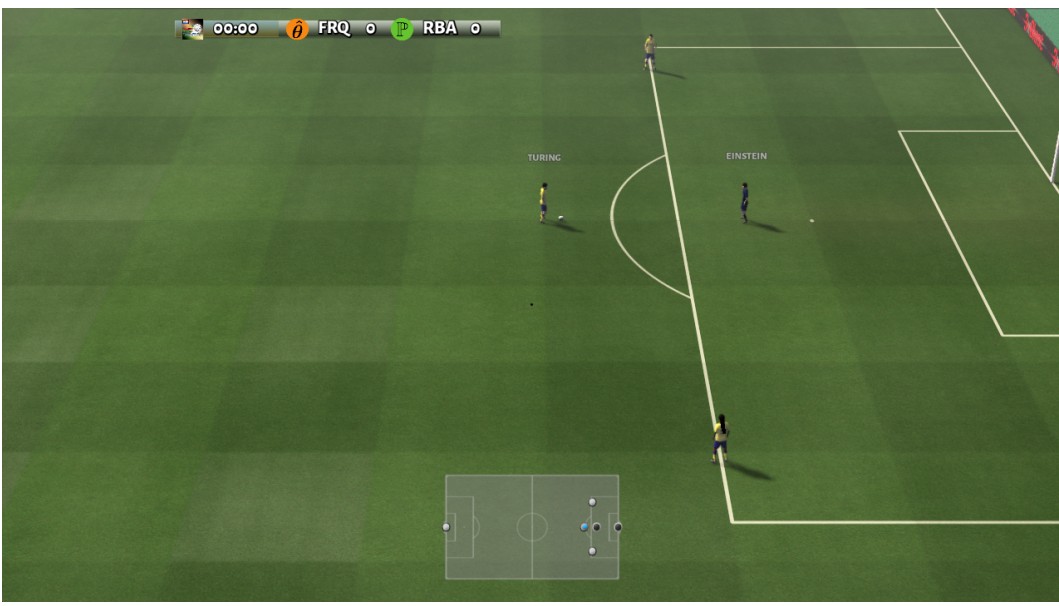

Figure 9: Google Research Football Academy 3 vs. 1 with Keeper scenario.

## C  Experiments Details

### C.1  Evaluation Process

We implement six ZSC algorithms including FCP [51], MEP [65], TrajeDi [30] , COLE [23], HSP [60] and E3T [58]. And we also implement self-play [5] as a baseline for comparison.

For Overcooked environment, we evaluate the ZSC capability by the BR-Prox metric across five random seeds. For each seed, we evaluate the ego agent with 30 evaluation partners for the Asymm. Adv. and Asymm. Coord. layouts and 20 evaluation partners for other layouts, and for 50 episodes each partner.

For GRF environment, we evaluate the ZSC capability by the BR-Prox metric across three random seeds due to the computational resource limitation. For each seed, we evaluate the ego agent with six evaluation partners with 168 combinations of partners. Each combination of partners has 10 episodes, and a total of 1680 episodes for each partner.

### C.2  ZSC algorithms Introduction

**Self-play.** Self-play (SP) [53] is a general approach in reinforcement learning, where agents only learn through playing against themselves. While it can yield high returns during training, agents trained using this method often struggle to coordinate with diverse policies. We training $10,000,000$ steps for SP agents.

**FCP.** Fictitious Co-Play (FCP) [51] is a two-stage training framework. In the first stage, it creates a diverse partner population through self-play agents pre-trained with different seeds and their previous checkpoints. In the second stage, it iteratively trains an FCP agent by having it play against sampled partners from the population. For the co-play methods including FCP, MEP and TrajeDi, we train $5e7$, $8e7$, $1e8$ steps for population sizes of 12, 24, 36 respectively.

**MEP.** Maximum Entropy Population-based training (MEP) [65] is a variant of FCP. It adopts the maximum entropy as a population diversity bonus added to the task reward, which is used as the objective to train a maximum entropy population in the first stage. In the second stage, it trains an robust agent by prioritized sample agents from the population. We observe that $\beta$ for prioritized sampling should be small when the population size is large. Thus we use $\beta = 0.5$ in our experiments.

**TrajeDi.** Trajectory Diversity PBT (TrajeDi) [30] aims to improve the policy diversity by adding a diversity measure to PBT losses. In details, it introduces the Jensen-Shannon divergence to the loss when training the population. We implement TrajeDi as a two-stage algorithm. We first train a

population with the Jensen-Shannon divergence to encourage diversity and then train the ego agent with uniformly sampling the population. Due to the time consumption problem, we calculate the JSD by sampling the population instead of traversing the population.

**HSP.** Hidden-utility Self-Play (HSP) [60] constructs the training population is analogously to how we construct evaluation. HSP constructs a pool of behavior-preferring agents using event-based rewards and select half of them by greedy-selection. The population is then used to train the ego agent with a mixture of behavior-preferring and MEP-trained partners. The main difference in population construction is that we use BR-Div to select evaluation companions and restrict the event-based reward space in order to promote reseaonable behavior.

**E3T.** Efficient End-to-End Training (E3T) [58] employs a mixture of ego policy and random policy to construct the partner policy and trains the ego agent without the need of a pre-trained population. We implement E3T without the partner modeling module for a fair comparison. We select the balance parameter $\epsilon$ as $0.5$ and the decaying factor of neural network parameters $\alpha$ as $0.1$.

**COLE.** Cooperative Open-ended Learning (COLE) [23, 24] constructs open-ended objectives in two-player cooperative games from the perspective of graph theory. With the objective functions calculated using the cooperative incompatibility distribution, it approximates the local best-preferred strategy to expand the population, which overcomes the cooperative incompatibility problem disclosed by other approaches. We implement the mete-solver using a reward-based ranking instead of the Shapley Value due to the time consumption. We train 50, 100 and 150 generations for population size of 25, 50 and 75 respectively and train 1,000,000 steps for a generation.

### C.3 Important Implementation Details

We implement the main body of ZSC-Eval based on HSP's implementation [4] [60], the cooking simulation environment from Overcooked-AI[5][5], the football simulation environment from Google Research Football [20].[6] All our experiments were run on Linux servers including two types of nodes: 1) 1-GPU node with NVIDIA GeForce 3090Ti 24G as GPU and AMD EPYC 7H12 64-Core Processor as CPU, 2) 2-GPU node with two GeForce RTX 3090 24G as GPUs and AMD Ryzen Threadripper 3970X 32-Core Processor as CPU.

**Parallel Partner Sampling**. When training the PPO algorithm, we sampling the episodes in which the ego agent plays with different partners in a batch, which makes the training framework more scalable.

**Centralized Critic**. Recent works have verified that a centralized critic function benefits the performance in fully cooperative games [59, 55, 44].

**Truncated Infinite Game**. As emphasized in Gymnasium[7], Kostrikov and Raayai Ardakani [18] and Pardo et al. [39], wrong calculation of the truncated returns leads may break the MDP properties of the environments. We choose to discard the value function iteration from the truncated states.

**Available Actions**. We implement basic available action indications in the Overcooked and GRF environments, such as avoiding keeping hitting the counter and null interaction, to accelerate the exploration.

**Entropy Coefficients Decay**. To encourage discovering more high-performing coordination conventions, we choose to use large entropy coefficients and decay the entropy coefficients during training. The linear entropy coefficients decay mechanism is summaried in Tables 5 and 6.

**Population Size**. In Overcooked, we choose the population size as 12, 24 and 36 for the co-play methods to demonstrate the effects of population size. While choose the population size as 25, 50, 75 for COLE since the evolution methods generate the ego agent end-to-end without pre-trained populations and thus require large populations to achieve better performance. In GRF, we choose the population size as 9 for the co-play methods and 15 for COLE due to the limit of computation resources.

---

[4] https://github.com/samjia2000/HSP, with with MIT License.

[5] https://github.com/HumanCompatibleAI/overcooked_ai, with MIT license.

[6] https://github.com/google-research/football, with MIT license.

[7] https://gymnasium.farama.org/tutorials/gymnasium_basics/handling_time_limits.

Table 5: Entropy coefficient schedulers in Overcooked.

| Method | Population Size | Entropy Coefficient Schedules | Entropy Coefficient Milestones |
|---|---|---|---|
| Co-play | 12 | 0.2 0.05 0.01 | 0 $2.5e7$ $5e7$ |
| | 24 | 0.2 0.05 0.01 | 0 $4e7$ $8e7$ |
| | 36 | 0.2 0.05 0.01 | 0 $5e7$ $10e7$ |
| Evolution | 25 | 0.2 0.05 0.01 | 0 $2.5e7$ $5e7$ |
| | 50 | 0.2 0.05 0.01 | 0 $5e7$ $10e7$ |
| | 75 | 0.2 0.05 0.01 | 0 $7.5e7$ $1.5e8$ |

Table 6: Entropy coefficient schedulers in GRF.

| Method | Population Size | Entropy Coefficient Schedules | Entropy Coefficient Milestones |
|---|---|---|---|
| Co-play | 9 | 0.2 0.01 0.01 | 0 $1.5e7$ $3e7$ |
| Evolution | 15 | 0.02 0.01 0.01 | 0 $1.8e7$ $3.6e7$ |

**Important Hyperparameters**. We use mostly the same hyperparamters as in Yu et al. [60], except for the mentioned details such as the entropy coefficients.

**Event-based Reward Space Design and Policy Behavior Feature**. We design a set of events and their corresponding range of weights, as summarized in Tables 7 and 8. Using $B_{\max} = 20$ and $C_{\max} = 3$, we generate up to 194 candidates and select up to 30 evaluation partners. The generated candidates are excluded if they cannot complete a delivery when cooperating with their BRs. The behavior feature of a policy is embedded as the occurrence of these events during the episodes.

Table 7: Designed events and weights used in Overcooked.

| Events | Weights |
|---|---|
| Put an onion or a dish or a soup onto the counter | 0 |
| Pickup an onion or a dish or a soup from the counter | 0 |
| Pickup an onion from the onion dispenser | -20,0,10 |
| Pickup a dish from the dish dispenser | -20,0,10 |
| Pickup a soup | -20,0,5,10 |
| Place an ingredient into the pot | -20,0,3,10 |
| Deliver a soup | -20,0 |
| Stay | -0.1,0,0.1 |
| Movement | 0 |
| Order Reward | 0.1,1 |

Table 8: Designed events and weights used in GRF.

| Events | Weights |
|--------|---------|
| Pass | -5,0,1 |
| Catch | -5,0,1 |
| Shot | -5,0,1 |
| Assist | 0 |
| Possession | 0 |
| Goal Reward | 1,5 |

## C.4 Human Experiment Details

### C.4.1 Experiment Setup

We recruited participants ($N = 152$) using an internal university platform, and we verified 145 valid data points. These participants were aged between 18 and 35, with a gender distribution of 90 for males and 55 for females. 70 participants have experience in playing the real Overcooked! game. Using a within-subjects experimental design, each participant engaged in experiments with 7 different agents across 2 different layouts, resulting in a total of 14 rounds. To mitigate learning effects among the subjects, both the order of the layouts and the agents were randomized. Each game round lasted for 400 time steps (approximately one minute). Each participant earns RMB 58.79 Yuan for the experiment. The names of the algorithms used by the agents were not visible during the experiments; instead, colors were used for differentiation. Participants were asked to rank the agents on the same layout after each round. We also recorded the scores and trajectories of each round. All data collection was conducted with the consent of the participants. After data collection, the data were de-identified, removing all personally identifiable information.

### C.4.2 Experiment Platform

We implement our human experiment platform based on the COLE-Platform [24].[8] The experimental platform is shown in Figures 10 to 13.

---

[8]`https://github.com/liyang619/COLE-Platform`, with MIT License.

# Experimental Statement

## 1. Purpose

You have been asked to participate in a research study that studies human-AI coordination. We would like your permission to enroll you as a participant in this research study.

The instruments involved in the experiment are a computer screen and a keyboard. The experimental task consisted of playing the computer game Overcooked and manipulating the keyboard to coordinate with the AI agent to cook and serve dishes.

## 2. Procedure

In this study, you should read the experimental instructions and ensure that you understand the experimental content. The whole experiment process lasts about **30** minutes, and the experiment is divided into the following steps:

(1) Read and sign the experimental statement, and you need to fill in a questionnaire;

(2) Test the experimental instrument, and adjust the seat height, sitting posture, and the distance between your eyes and the screen. Please ensure that you are in a comfortable sitting position during the experiment;

(3) You will first try out the game actions you learned in the tutorial within a simple layout to familiarize yourself with the game mechanics;

(4) Start the formal experiment. Please cooperate with the AI agents to get as much scores as possible. You will play with 7 agents in each layout and total 2 layouts. You need to rank the performance of these seven agents in each layout. After each round, we will ask you to add the current agent to the ranking. After the game ends in each layout, we need to confirm your ranking of the agents.

## 3. Risks and Discomforts

The only potential risk factor for this experiment is trace electron radiation from the computer. Relevant studies have shown that radiation from computers and related peripherals will not cause harm to the human body.

## 4. Costs

Each participant who completes the experiment and fills correct individual information will be paid 50 ~ 65 RMB according to your performance.

## 5. Confidentiality

The results of this study may be published in an academic journal/book or used for teaching purposes. However, your name or other identifiers will not be used in any publication or teaching materials without your specific permission. In addition, if photographs, audio tapes or videotapes were taken during the study that would identify you, then you must give special permission for their use.

I confirm that the purpose of the research, the study procedures and the possible risks and discomforts as well as potential benefits that I may experience have been explained to me. All my questions have been satisfactorily answered. I have read this consent form. Clicking the button below indicates my willingness to participate in this study

# 实验说明

## 目的

您被邀请参与一项研究人类与人工智能协作的研究。我们希望您能同意作为参与者加入这项研究。

实验中使用的设备包括电脑屏幕和键盘。实验任务是玩电脑游戏《胡闹厨房》，并使用键盘与AI代理协作烹饪和上菜。

## 实验流程

在本研究中，您需要阅读实验说明并确保理解实验内容。整个实验过程约持续30分钟，实验分为以下步骤：

(1) 阅读并签署实验说明，并填写人口统计信息；

(2) 测试实验设备，调整座椅高度、坐姿以及眼睛与屏幕之间的距离。请确保在实验过程中保持舒适的坐姿；

(3) 您将首先在一个简单布局中尝试您在教程中了解的游戏操作，并熟悉游戏机制；

(4) 开始正式实验。请与AI代理合作，尽可能获得高分。您将在每个布局中与7个代理进行游戏，共2个布局。您需要对这七个代理在每个布局中的表现进行排序。每轮结束后，我们将请您将当前代理加入排序。每个布局的游戏结束后，我们需要确认您对代理的排序。

## 风险和不适

本实验唯一可能的风险因素是来自计算机的微量电子辐射。相关研究表明，计算机及相关外设的辐射不会对人体造成伤害。

## 报酬

每位完成实验并正确填写个人信息的参与者将根据表现获得50至65元人民币的报酬。

## 实验数据保密性

本研究的结果可能会发表在学术期刊书籍中或用于教学目的。然而，未经您的具体许可，您的姓名或其他识别信息不会在任何出版物或教学材料中使用。此外，如果在研究过程中拍摄了可能识别出您的照片、录音或录像，则必须获得您的特别许可可能使用。

我确认研究目的、研究程序、可能的风险和不适以及我可能体验到的潜在益处已向我解释清楚。我所有的问题都已得到满意的回答。我已阅读本同意书。点击下面的按钮表示我愿意参与本研究

I have read and agreed all the experimental statement above. Start experiment.

Figure 10: Statement

## Instructions

**Please read the following instructions carefully.**

In this task, you will play in a cooking game as one of the two chefs in a restaurant that serves onion soup. The chef in you control wearing a **gray hat**.

One of the game layouts looks like:

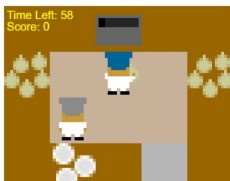

There are a number of objects in the game, labeled here:

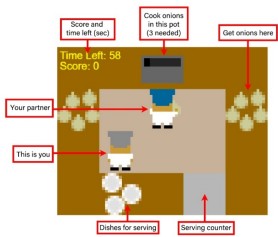

### Movement and interactions

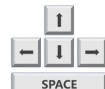

You can move up, down, left, and right using the **arrow keys**, and interact with objects using the **spacebar**.

You can interact with objects by facing them and pressing **space bar**.

Note that you and your partner **cannot occupy the same location**.

### Cooking

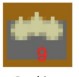 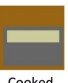

Cooking     Cooked
Soup         Soup

Once 3 onions are in the pot, the soup begins to cook. After the timer gets to 20, the soup will be ready to be served. To serve the soup, bring a dish over and interact with the pot.

### Goal

Your goal in this task is to serve as many of the orders as you can before each level ends. The current score and time left for you are shown in the upper left of game.

---

**After clicking "Start Playing", you will first play in a warmup trial, where scores will not be recorded.**

**After the warmup trial, the official experiments will be conducted in 3 layouts. You will complete 7 games with 7 different agents in each layout.**

---

**When playing, please make sure to pay attention to the hat colors of your partners! You will be asked to rank the agents (identified by their hat colors) according to their performance.**

**在游戏中，请确保注意你伙伴的帽子颜色！你将需要根据他们的表现，按帽子颜色对智能体进行排序。**

Start Playing

Figure 11: Experiment Instruction

| Layout | Game Length (sec) | Game number | Agent ID |
|--------|-------------------|-------------|----------|
| random3 | 75 | 1 of 14 | 1 of 7 |

When playing, please make sure to pay attention to the hat colors of your partners! You will be asked to rank the agents (identified by their hat colors) according to their performance.

在游戏中，请确保注意你伙伴的帽子颜色！你将需要根据他们的表现，按帽子颜色对智能体进行排序。

You will get another 5 Yuan if you achieve a score of 160 or more with any agent partner in this map layout.

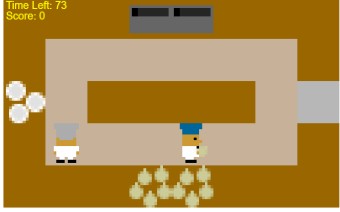

Figure 12: Main Experiment

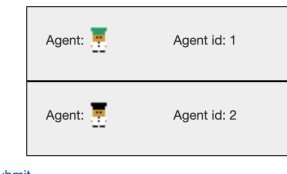

## Questionnaire

Please rank the agents by dragging the corresponding figures based on your feelings of the agents' cooperation ability.

Please rank the agents **from best to worst, from top to bottom.**

请根据你对智能体合作能力的感觉，通过拖动相应的图形来对智能体进行排序。

请将智能体按照表现从好到坏，从上到下排序。

| | |
|---|---|
| Agent: | Agent id: 1 |
| Agent: | Agent id: 2 |

Submit

Figure 13: Human Subjective Perception Ranking

# D    Additional Results

## D.1    Details of Figure 2(b)

We trained 176 evaluation partner candidates in Overcooked Coord. Ring layout, then select subsets according to BR-Div and P-Div, with subset size ranging from 2 to 15. We remark that only the comparisons with the same subset sizes are meaningful. We guess that the population diversity of the two methods first increases and then decreases because possibly correlated partners are included and that 0 values mean that linearly correlated partners are included.

## D.2    Visualization of high-level behaviors

In Figure 14, we visualize the statistic data of the high-level events (Section 4.1) collected in Overcooked through principal components analysis [10]. As is shown in the visualization result:

- Populations trained using the MEP [65] method differ in random initialization seeds, hyper-parameters and network architectures but learn similar behaviors.
- Evaluation partners selected by maximizing BR-Div exhibit the most diverse behaviors, while maximizing P-Div leads to inferior behavior discovering.

Figures 17, 18 and 19 show the heatmap of the evaluation partners' high-level behaviors in Coordination Ring, Asymmetric Coordination and GRF.

## D.3    Diversity of Evaluation Partners in Episode Returns

We illustrate that our evaluation partners are more comprehensive for ZSC capability evaluation than the evaluation partners generated by previous evaluation methods in Overcooked Asymmetric Advantages layout, as shown in Figure 15.

## D.4    Analyzing Training Population with BR-Div

The proposed BR-Div can also be an analysis tool for the effectiveness of ZSC algorithms generating training population. According to Figures 4 and 6, performance can be enhanced by increasing population size, provided that the population also increases in diversity, i.e., the population diversity is effectively enlarged by ZSC algorithms. Some ZSC algorithms lack an explicit mechanism to promote the population diversity, including FCP and COLE. Thus the performance of FCP and COLE is not benefited from increasing the population size from 24 to 36.

## D.5    Overcooked

Figures 22 and 23 show the performance of ZSC algorithms in all the 7 layouts. The black line marked on each bar is the interquartile mean of the data.

Table 9: Percentage of Ranks.

| % \ Rank Algo | 1 | 2 | 3 | 4 | 5 | 6 | 7 |
|---|---|---|---|---|---|---|---|
| SP | 0.0 | 0.0 | 3.7 | 7.41 | 3.7 | 3.7 | 81.48 |
| E3T | 3.7 | 14.81 | 0.0 | 11.11 | 37.04 | 33.33 | 0.0 |
| FCP | 3.7 | 7.41 | 44.44 | 11.11 | 14.81 | 14.81 | 3.7 |
| MEP | 29.63 | 37.04 | 11.11 | 11.11 | 3.7 | 7.41 | 0.0 |
| TrajDi | 7.41 | 11.11 | 18.52 | 37.04 | 11.11 | 3.7 | 11.11 |
| COLE | 7.41 | 7.41 | 11.11 | 7.41 | 25.93 | 37.04 | 3.7 |
| HSP | 48.15 | 22.22 | 11.11 | 14.81 | 3.7 | 0.0 | 0.0 |

Table 9 and Figures 20 and 21 summarize the performance rank under BR-Prox with 3 different population sizes.

Figure 24 shows the methods' performance with different skill level evaluation partners in three 'coordination with conflicts' layouts and three Multi-recipe layouts.

### D.6 Google Research Football

We show the performance in episode returns (goal scores) in Table 3.

Table 10: Return performance with 95% confidence intervals of ZSC algorithms in Google Research Football.

| Method | Goal | (95% CI) |
|---|---|---|
| SP | 0.09 | (0.07, 0.12) |
| E3T | 0.31 | (0.29, 0.37) |
| FCP | 0.38 | (0.35, 0.43) |
| MEP | 0.38 | (0.36, 0.42) |
| TrajeDi | **0.40** | **(0.37, 0.45)** |
| COLE | 0.36 | (0.33, 0.42) |
| HSP | **0.40** | **(0.36, 0.44)** |

As shown in Table 11, we conduct a ablation study of E3T [58] in the GRF environment.

Table 11: Ablation study of E3T: Balance Parameter.

| | 0.05 | 0.1 | 0.25 | 0.5 |
|---|---|---|---|---|
| **Goal(95% CI)** | 0.28 (0.26,0.32) | 0.30 (0.26,0.34) | 0.29 (0.25,0.32) | 0.32 (0.29,0.37) |
| **BR-Prox (95% CI)** | 0.60 (0.53,0.66) | 0.60 (0.53,0.67) | 0.58 (0.51,0.65) | 0.66 (0.59,0.73) |

### D.7 Human Experiment

We illustrate the objective ranks and subjective ranks on different ZSC algorithms in Figures 25 and 26, which are consistent.

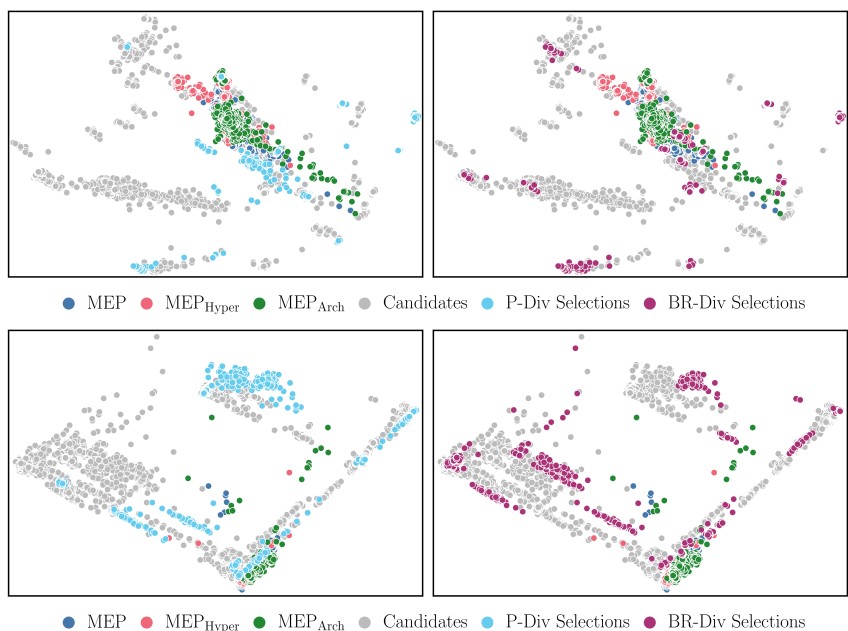

Figure 14: Visualization of the high-level behaviors of different self-play populations and our evaluation partner candidates. The evaluation partners selections are sampled according to partner diversity and BR-Div respectively. **ToP:** Counter Circuit. **Bottom:** Asymmetric Advantages.

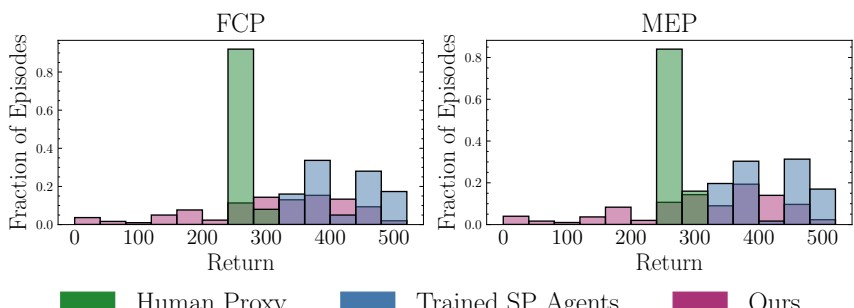

Figure 15: Distributions of episode returns computed by evaluating FCP and MEP ZSC agents with the our evaluation partners and trained self-play agents.

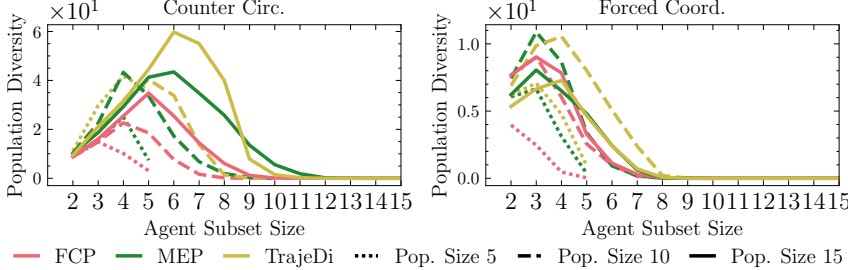

Figure 16: Effect of the population size on the population diversity. Near 0 values mean that linearly correlated evaluation partners are included.

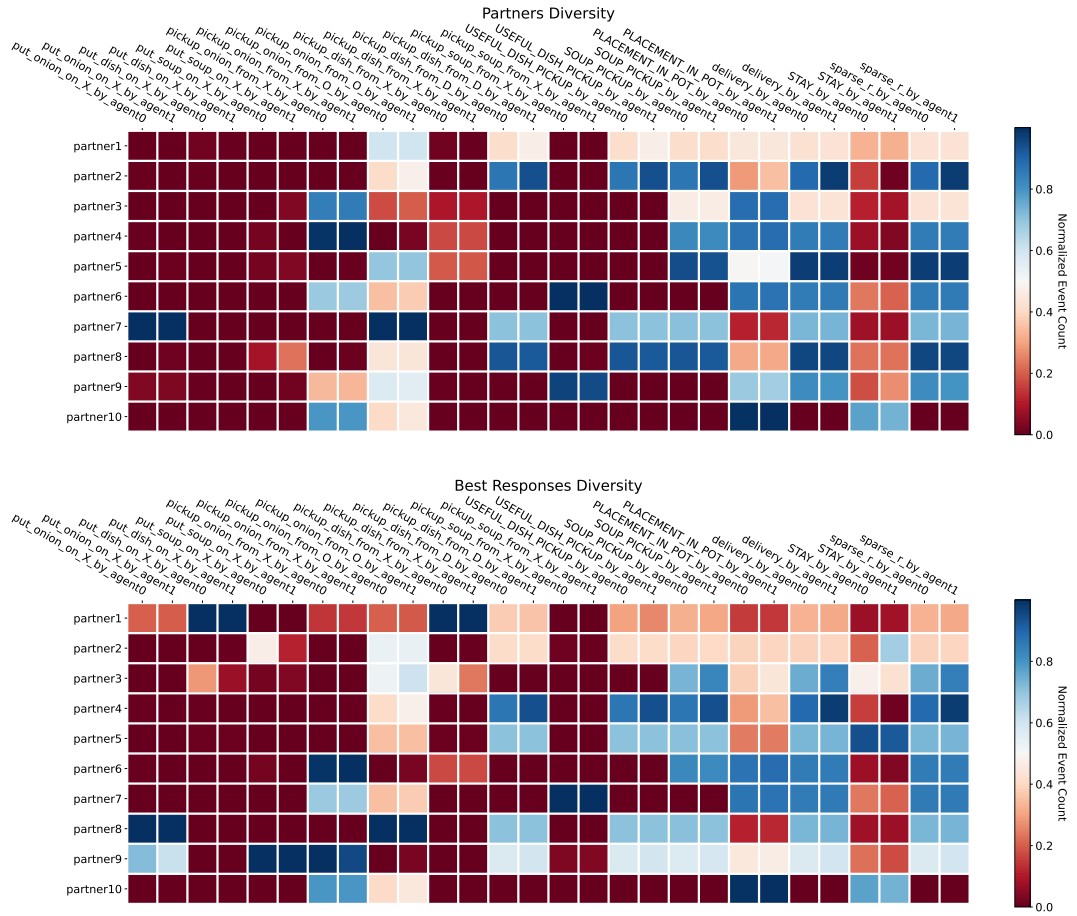

Figure 17: Heatmap of the evaluation partners' high-level behaviors of the Coord. Ring scenario in the Overcooked Environment. The BR-based Diversity maximization produces evaluation partners that use the counter more frequently.

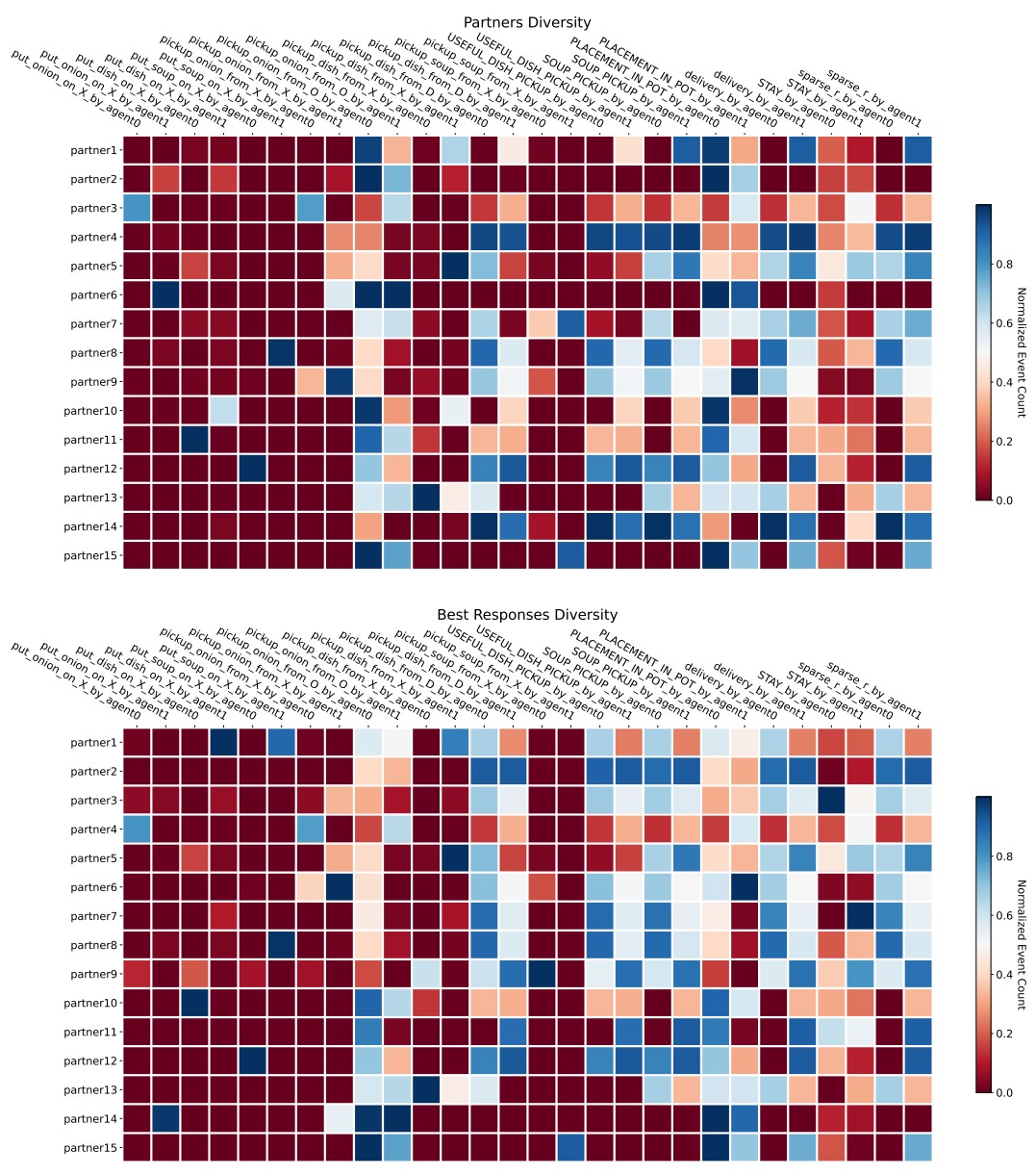

Figure 18: Heatmap of the high-level behaviors of the Asymm. Coord. scenario in the Overcooked Environment. The BR-based Diversity maximization produces evaluation partners that use the counter more frequently and deliver the soup in both sides.

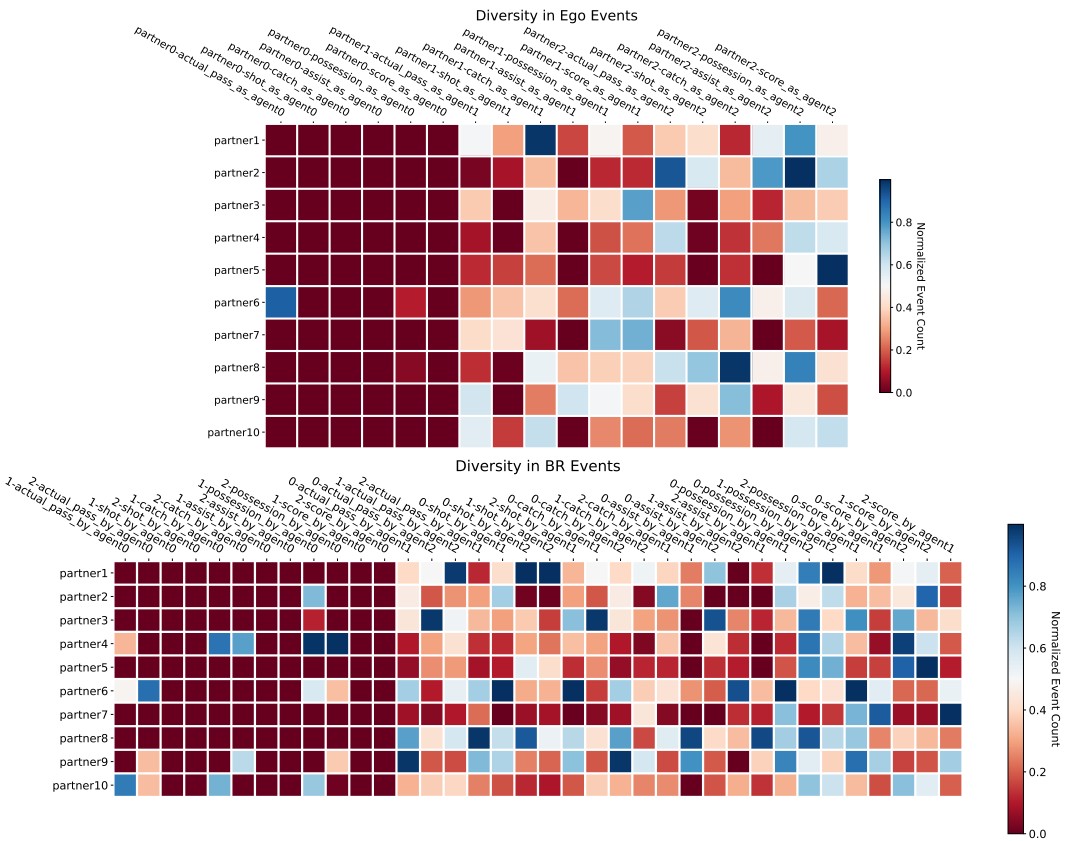

Figure 19: Heatmap of the high-level behaviors of the Academy 3 vs. 1 with keeper scenario in the Google Research Football environment. Our evaluation workflow generates partners with diverse results in both partners' behaviors and BRs' behaviors.

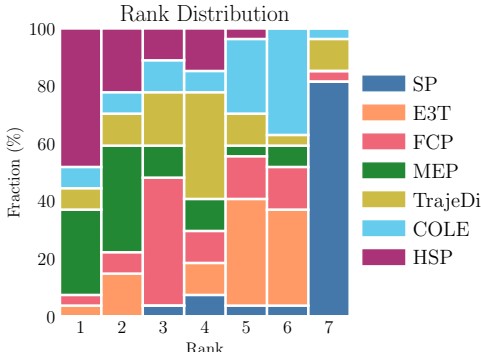

Figure 20: Rank of different ZSC algorithms.

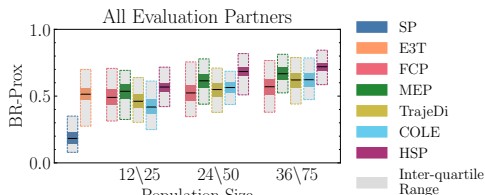

Figure 21: BR-Prox performance of different ZSC algorithms in all 'coordination with conflicts' layouts and layouts with multiple recipes.

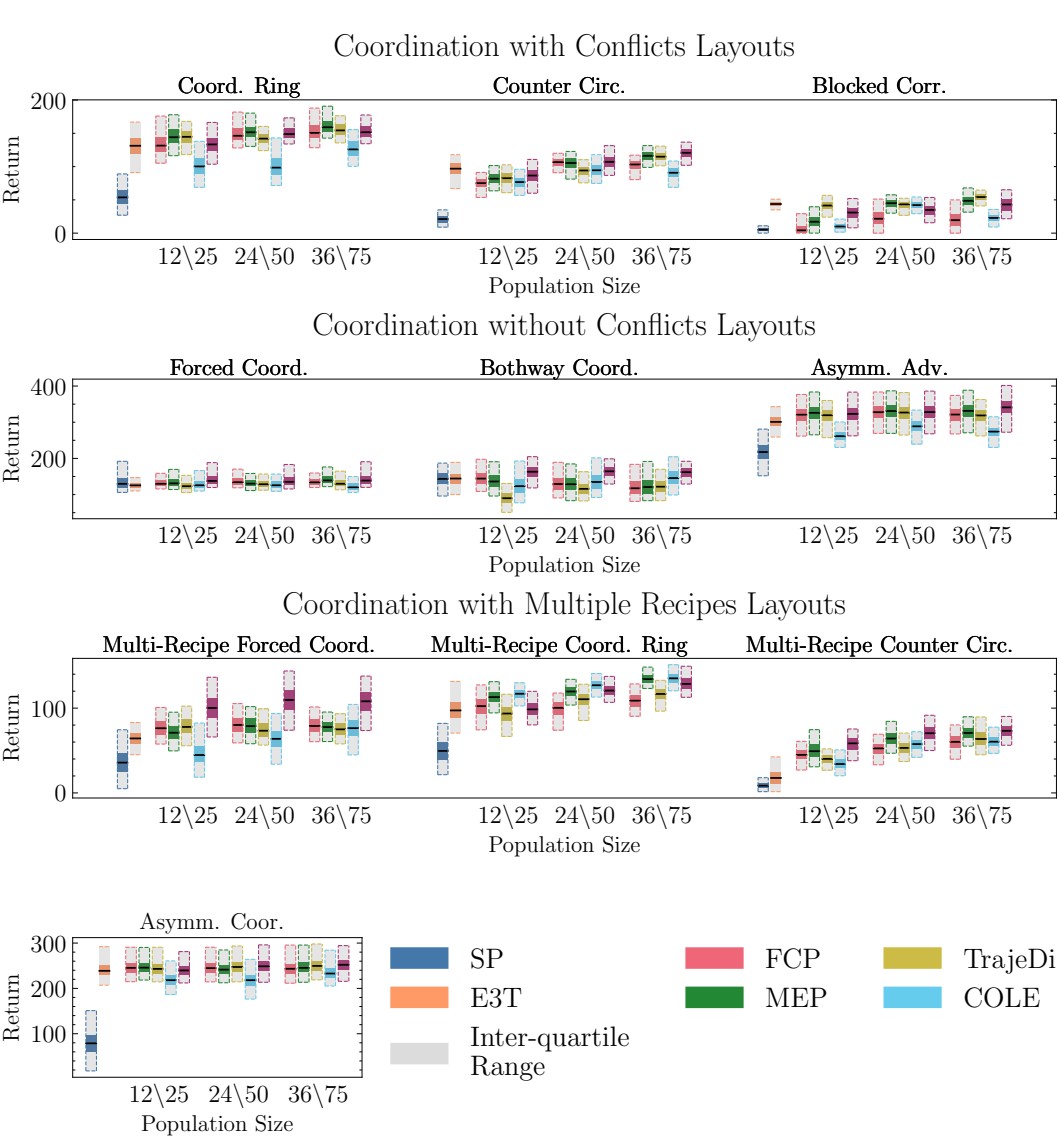

Figure 22: Episode return performance with 95% confidence intervals of ZSC algorithms with different population sizes in the Overcooked environment.

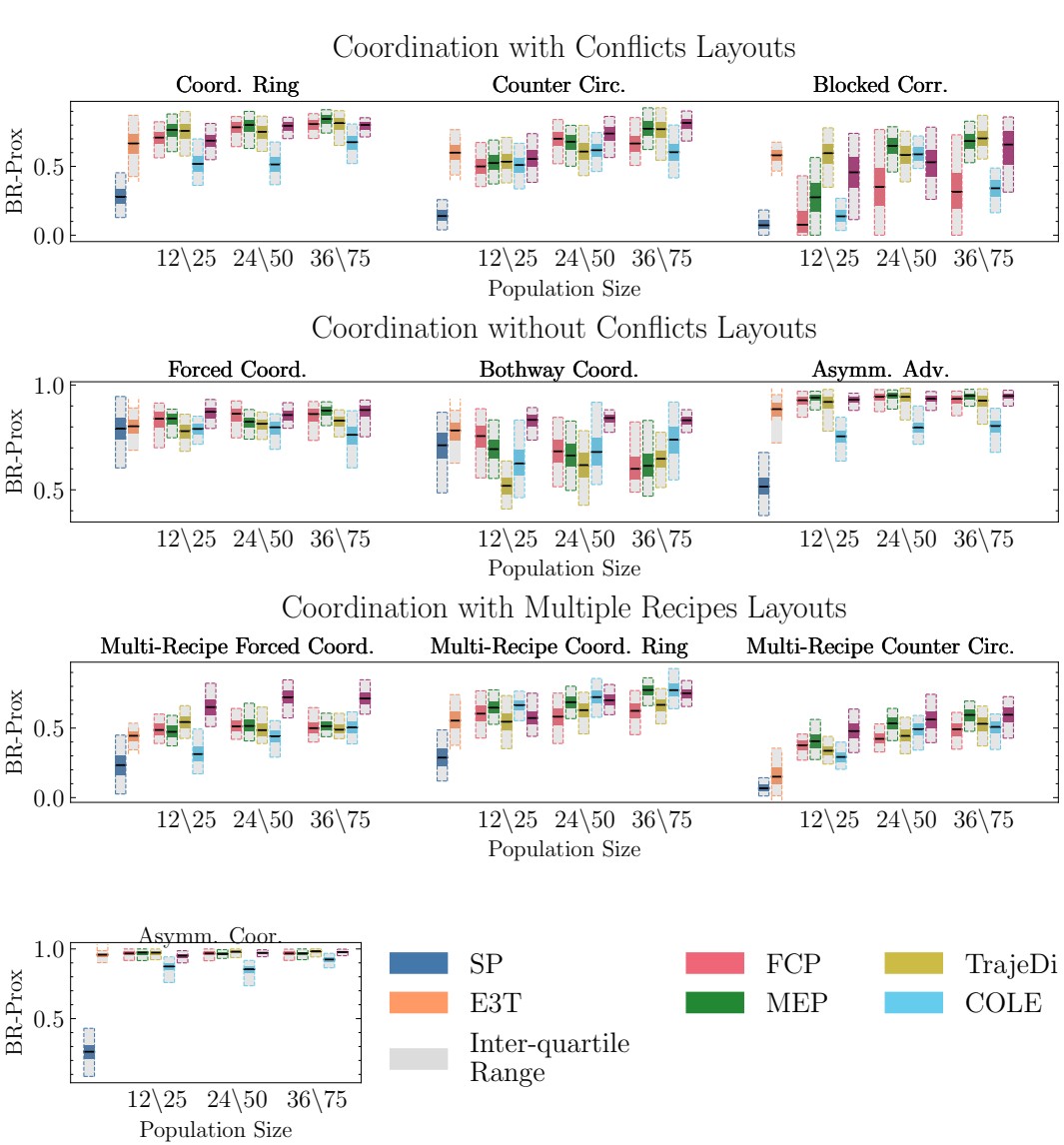

Figure 23: BR-Prox performance with 95% confidence intervals of ZSC algorithms with different population sizes in the Overcooked environment.

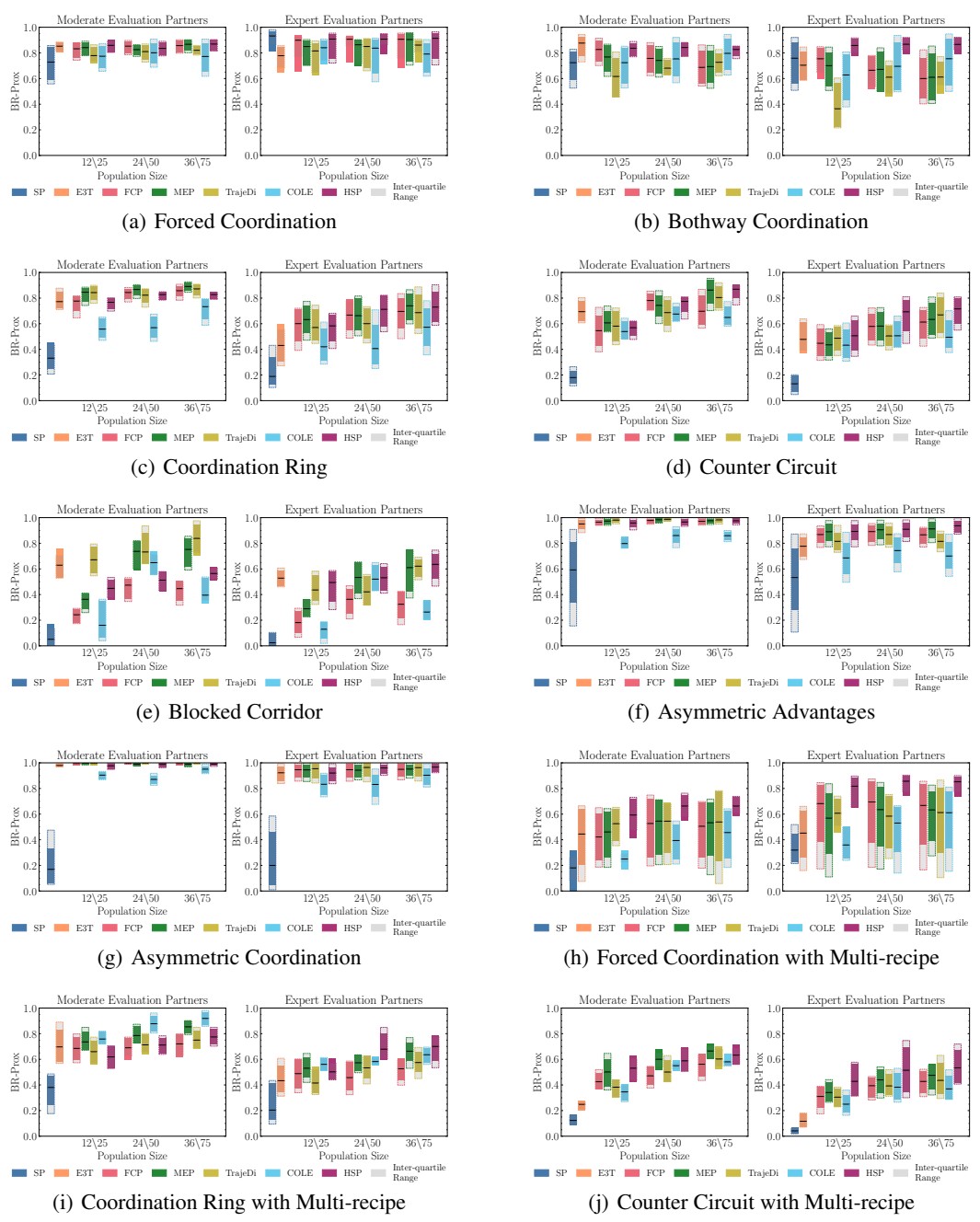

Figure 24: BR-Prox performance with 95% confidence intervals obtained with evaluation partners at different skill levels.

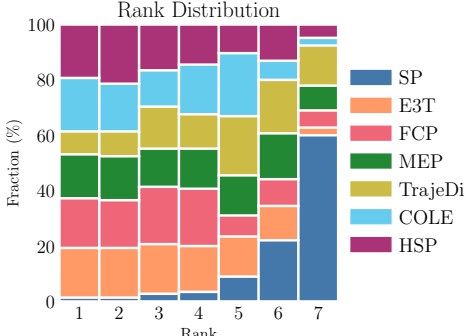

Figure 25: Human subjective feelings ranks on different ZSC algorithms.

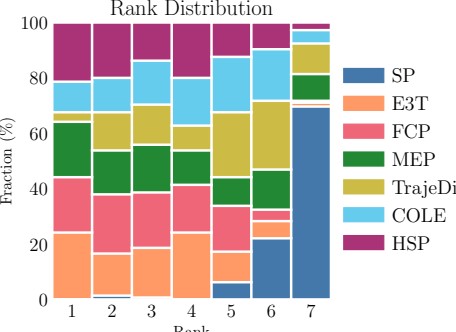

Figure 26: Episode returns ranks on different ZSC algorithms.

