# OpenReview forum: "ZSC-Eval: An Evaluation Toolkit and Benchmark for Multi-agent Zero-shot Coordination"
_NeurIPS.cc/2024/Datasets_and_Benchmarks_Track — NeurIPS 2024 Track Datasets and Benchmarks Poster_

### Official Review · Reviewer_9xeX · 2024-07-23

**Rating:** 6
**Confidence:** 2
**Correctness:** Good
**Clarity:** Good

**Review:**

Pros

- ZSC-Eval provides a robust framework for evaluating ZSC algorithms, which includes partner generation, selection, and performance measurement.
- The toolkit generates evaluation partners that exhibit diverse high-level behaviors, addressing a significant limitation of existing methods that often produce similar training and evaluation partners.
- The toolkit is demonstrated to scale effectively to more complex scenarios, such as those in Google Research Football, making it versatile for various environments.
- ZSC-Eval introduces the Best Response Proximity metric, which better captures generalization performance and balances results based on partners' cooperation capabilities.
- The experiments conducted using ZSC-Eval reveal new insights and provide guidelines for designing ZSC testbeds, enhancing the understanding of ZSC capabilities and limitations.

Cons

- The writing is not easy to follow. What does it mean by $\pi^{-i}$ in line 103? Could you please explain more about equations in lines 107 & 110?
- The effectiveness of ZSC-Eval has been demonstrated primarily in specific environments like Overcooked and GRF, which might limit its generalizability to other domains.

**Strengths:**

See pros

**Additional Feedback:**

See review

**Documentation:**

Good

**Limitations:**

The effectiveness of ZSC-Eval has been demonstrated primarily in specific environments like Overcooked and GRF, which might limit its generalizability to other domains.

**Opportunities For Improvement:**

See Cons

**Relation To Prior Work:**

Good. The submission clearly discussed how this work differs from previous work.

**Summary And Contributions:**

The submission introduces ZSC-Eval, a comprehensive toolkit and benchmark designed to evaluate ZSC algorithms in multi-agent RL. The contributions of this work include the generation of evaluation partners via behavior-preferring rewards, selection of partners using Best-Response Diversity, and measurement of ZSC capability through the Best-Response Proximity metric, providing a more robust and practical evaluation framework compared to existing methods .

---

> ### Author Rebuttal · Authors · 2024-08-16
>
> We thank the reviewer for the insightful suggestions and recognition of our work.
> For your suggestions and problems, we reply point by point:
>
> >Q: **What does it mean by $\pi^{-i}$ in line 103? Could you please explain more about equations in lines 107 & 110?**
>
> A: Thank you for pointing out the issue. We sincerely apologize for not clearly explaining the meaning of $\pi^{-i}$. $\pi^{-i}$ usually represents the joint policy of the agents in $\mathcal{N}$ except the ego agent, which is usually marked as agent $i$, while we use $\pi^{-i}$ in a scenario where there are multiple agents controlled by policy $i$, i.e., the policy of agent $i$. Therefore $\pi^{-i}$ means the joint policy of the agents in $\mathcal{N}$ except those controlled by the ego agent's policy.
>
> The equation in Line 107 is the optimization objective of the ego agent, which aims at maximizing the expected performance with all possible unseen partners. $\mathcal{U}$ is a uniform distribution and $\Pi_{test}$ is the set of unseen partners. Since there may be more than 2 agents in a scenario, we should consider the possible combinations of different unseen partners, we use $\mathcal{P}(\Pi_{test})$ to denote those combinations. Put the things together, $\mathbb{L}\sim\mathcal{U}(\mathcal{P}(\Pi_{test}))$ is a combination of unseen partners sampled following a unified distribution, and $\mathcal{J}(\pi, \{\pi^{i}\}_{i\in\mathbb{L}})$ is the evaluation performance of the ego agent and combination $\mathbb{L}$. Finally, $\max _{\pi}\mathbb{E} _{\mathbb{L}\sim\mathcal{U}(\mathcal{P}(\Pi _{test}))}[\mathcal{J}(\pi, \{\pi^{i}\} _{i\in\mathbb{L}})]$ means maximizing the expected evaluation performance of the ego agent and any combination of the unseen partners sampled uniformly.
>
> The equation in Line 107 does not include the training procedure. We focus on population-based ZSC algorithms, so we denote the population of training agents as $\Pi_{train}$, and the training procedure as $\mathcal{O}$, which computes the common best response to the partners in $\Pi_{train}$, e.g., a RL policy that is trained pairing with $\Pi_{train}$. Put all the things together, the equation in Line 110 is the training objective of population-based ZSC algorithms: constructing a training population and developing a training procedure for maximizing the expected evaluation performance of the ego agent and any combination of the unseen partners sampled uniformly.
>
> We will make the necessary modifications in the future version to improve and make it clearer.
>
> >Q: **The effectiveness of ZSC-Eval has been demonstrated primarily in specific environments like Overcooked and GRF, which might limit its generalizability to other domains.**
>
> A: We need to claim that both Overcooked and GRF environments we used to verify the effectiveness of ZSC-Eval are the most mainstream and representative environments in ZSC research.
>
> Secondly, the key to generalizing our method to other environments lies in designing event-based rewards for those environments. Event-based rewards have been used in lots of complex tasks [1], including robotics [2], large-scale games [3] and product management [4]. It is relatively simple to implement in a well-studied environment where some reward and feature engineering has been completed.
>
> Furthermore, as formatted in Section 3.1, our ZSC-Eval principally can be extended to scenarios with a large number of agents. Empirically, in scenarios with a large number of agents, our ZSC-Eval framework can generate evaluation partners as long as the approximate NEs can be reached by the underlining MARL algorithms, e.g., MAPPO [5]. In other words, the effectiveness of our framework is not affected by the number of agents.
>
> Therefore, ZSC-Eval can be easy to generalize to other environments. Since more environments can benefit the research of zero-shot coordination, we are happy to integrate more environments and tasks. We will continuously develop our code repository to include more algorithms, environments and tasks.
>
>
> **If you feel that our responses have addressed your concerns and that our paper is of high quality, we would greatly appreciate your consideration of raising the score to a positive value.
> If you have any further questions or require additional information, please let us know.**
>
> Reference:
>
> [1] Fu, Justin, et al. "Variational inverse control with events: A general framework for data-driven reward definition." *Advances in neural information processing systems* 31 (2018).
>
> [2] Brohan, Anthony, et al. "Do as i can, not as i say: Grounding language in robotic affordances." *Conference on robot learning*. PMLR, 2023.
>
> [3] Ye, Deheng, et al. "Mastering complex control in moba games with deep reinforcement learning." *Proceedings of the AAAI Conference on Artificial Intelligence*. Vol. 34. No. 04. 2020.
>
> [4] Shi, Zhenyu, et al. "Learning expensive coordination: An event-based deep RL approach."*International Conference on Learning Representations*. 2020.
>
> [5] Yu, Chao, et al. "The surprising effectiveness of ppo in cooperative multi-agent games." *Advances in Neural Information Processing Systems* 35 (2022): 24611-24624.

---

> > ### Comment · Reviewer_9xeX · 2024-08-22
> >
> > Thanks to the author for the detailed response. My concerns were addressed. I increased my rating.

---

> > > ### Author Rebuttal · Authors · 2024-08-22
> > >
> > > Thank you very much for your positive feedback and for increasing your score. We will make the formulas clearer in the next version of the paper.
> > >
> > > Best regards,
> > > Authors of Submission 765

---

### Official Review · Reviewer_jqjf · 2024-07-25
**ZSC-Eval: An Evaluation Toolkit and Benchmark for Multi-agent Zero-shot Coordination**

**Rating:** 7
**Confidence:** 3
**Correctness:** Yes.
**Clarity:** Yes.

**Review:**

This paper presents a comprehensive toolkit for evaluating Zero-Shot Coordination (ZSC) in Multi-Agent Reinforcement Learning (MARL), marking the first rigorous evaluation framework specifically for ZSC. The work is of high quality, introducing innovative metrics such as Best-Response Diversity (BR-Div) and Best-Response Proximity (BR-Prox). However, the empirical validation could benefit from testing in a wider variety of environments to enhance its robustness.

**Pros** (Further details in the **Strengths** section):

* The paper introduces a new evaluation toolkit for ZSC with novel metrics and new scenarios in Overcooked.
* A full benchmark using the toolkit of well-known ZSC algorithms is provided.
* The paper is grounded in strong theoretical principles and experimental validation.

**Cons** (Further details in the **Opportunities for Improvement** section):

* Evaluations are consistent with human assessments; however, this validation is limited to experiments conducted in only two Overcooked scenarios.
* The diversity of environments tested is limited, reducing the generalizability of the findings.
* Figure 3 lacks quantitative metrics and detailed statistical analysis, providing only a qualitative illustration of behavioral diversity.
* Insufficient details are provided on the design and implementation of behavior-preferring rewards.
* The computational resources used for the experiments are not specified.
* The toolkit has been tested in a limited number of agent scenarios, which could constrain the applicability of the findings across different environments that require coordination of many agents.

**Strengths:**

* The paper is written well and conveys the main points clearly.
* This work is the first of its kind to develop a comprehensive evaluation and benchmark toolkit specifically for ZSC algorithms, addressing a significant gap in the MARL community.
* The authors not only utilized existing Overcooked scenarios but also created new ones, providing a richer and more varied testing ground for ZSC algorithms. This diversification of experiments offers valuable insights into the performance and limitations of current ZSC approaches.
* The paper is grounded in strong theoretical principles, which are justified through experimental validation.
* The paper introduces innovative metrics like Best-Response Diversity (BR-Div) and Best-Response Proximity (BR-Prox), which offer more nuanced and informative assessments of ZSC performance.

(edit: post rebuttal, score+1, 6->7)

**Additional Feedback:**

* The evaluation was conducted over 5 seeds in Overcooked and 3 seeds in GRF, which may be concerning as the literature suggests that rigorous evaluation should typically involve at least 10 seeds due to the non-deterministic behavior of RL algorithms.
* An in-depth analysis of failure cases where the ego agent struggles to coordinate with partners could provide insights into the limitations of the proposed methods and help improve future iterations.
* The new metrics, Best-Response Diversity (BR-Div) and Best-Response Proximity (BR-Prox), while innovative, lack extensive validation against established benchmarks or human intuition. Providing more empirical evidence would confirm their effectiveness and reliability.

**Documentation:**

The ZSC-Eval GitHub repository offers comprehensive instructions for generating evaluation partners, training ZSC methods, and conducting evaluations, ensuring reproducibility. However, it could benefit from adding detailed function-level documentation within the code.

**Ethics:**

No.

**Limitations:**

Yes, however further details could be beneficial.

**Opportunities For Improvement:**

Theoretically, the paper highlights the drawbacks and weaknesses of existing ZSC evaluation methods and shows the strength of the ZSC-Eval toolkit. However, empirically, some comparisons suffer from insufficient detail or lack extensiveness:

* Table 2 focuses on two Overcooked scenarios, which may not adequately represent the full range of coordination challenges, limiting the generalizability of the findings, it will be helpful to see such a comparison with additional scenarios and preferably, different environments beyond Overcooked.
* Figure 3 qualitatively illustrates the diversity of high-level behaviors among evaluation partners but lacks quantitative metrics and detailed statistical analysis, making it difficult to fully validate the effectiveness of ZSC-Eval.

The paper lacks sufficient details on certain aspects, such as the specific design and implementation of behavior-preferring rewards. Although Table 7 in the appendix lists events rewards chosen for the Overcooked scenarios (i.e., actions or sequences of actions performed by agents, such as picking up ingredients or serving a dish), it does not explain how to decide which events to include and how to assign their weights (the importance or influence of each event on the overall reward function). Therefore, to ensure effective coordination and high-quality evaluation partners, clear guidelines on choosing appropriate events and their corresponding weights for the behavior-preferring rewards would be beneficial.

The paper does not specify the compute resources used for the experiments. Including this information would be valuable for understanding the feasibility and reproducibility of the experiments, helping other researchers estimate the time and infrastructure required to replicate the results and validate the toolkit's scalability. Please provide details on the computational requirements and resources used.
With only Overcooked and the GRF '3 vs. 1 with Keeper' scenario as testbeds, there are still concerns about the limited number of agent scenarios used to test the toolkit. Adding a scenario with a larger number of evaluation partners alongside the ego agent would be helpful.

**Relation To Prior Work:**

Yes, the paper clearly discusses how this work differs from previous contributions, highlighting the limitations of existing ZSC evaluation methods and how ZSC-Eval addresses them.

**Summary And Contributions:**

This paper presents ZSC-Eval, a toolkit for evaluating Zero-Shot Coordination (ZSC) in Multi-Agent Reinforcement Learning (MARL).  Zero-Shot Coordination aims to train an “ego agent”, an agent that can be deployed to coordinate with unseen partners without additional training; these partners are called evaluation partners. ZSC-Eval generates diverse evaluation partners using "behavior-preferring rewards" and selects a representative subset via "Best-Response Diversity" (BR-Div). It then measures performance with "Best-Response Proximity" (BR-Prox), assessing how closely the ego agent approximates optimal responses. Evaluations in Overcooked and Google Research Football show consistent results with human evaluations, where human players act as evaluation partners to assess the coordination skills of the ego agent, and highlight issues like performance degradation with expert partners. ZSC-Eval offers a comprehensive evaluation framework and benchmarks, advancing the development of robust ZSC algorithms in the MARL community.

---

> ### Author Rebuttal · Authors · 2024-08-16
>
> We thank the reviewer for the insightful suggestions and recognition of our work.
> For your suggestions and problems, we reply point by point:
>
> >Q: **Human evaluation is limited to experiments conducted in only two Overcooked scenarios.**
>
> A: Given that our human experiment's main focus is to validate our framework's effectiveness in generating evaluation partners that can represent the deployment-time partners, we have already conducted a large-scale human evaluation experiment with **130 individuals** in the most popular ZSC environment Overcooekd. The current results have effectively validated the high consistency of our proposed method with real human evaluations. We believe these results support our claim that ZSC-Eval can serve as a low-cost alternative to humans as evaluation partners.
>
> Limited by the conditions for developing the experimental platform and the required funding, it is difficult for us to further expand the scale of the experiments. This work can be further explored in the future, and we consider the design of more human experiments in different environments and diverse comparison methods as important future works in this area.
>
> >Q: **The diversity of environments tested is limited, reducing the generalizability of the findings.**
>
> A: We need to claim that both the Overcooked and GRF environments we used to verify the effectiveness of ZSC-Eval are the most mainstream and representative environments in ZSC research.
>
> Secondly, the key to generalizing our method to other environments lies in designing event-based rewards for those environments. Event-based rewards have been used in lots of complex tasks [1], including robotics [2], large-scale games [3] and product management [4]. It is relatively simple to implement in a well-studied environment where some reward and feature engineering has been completed.
>
> Since adding more environments can benefit the research of zero-shot coordination, we are happy to integrate more environments and tasks. We will continuously develop our code repository to include more algorithms, environments, and tasks.
>
> >Q: **Figure 3 lacks quantitative metrics and detailed statistical analysis, providing only a qualitative illustration of behavioral diversity, making it difficult to fully validate the effectiveness of ZSC-Eval.**
>
> A: We first explain why we use qualitative illustration. We aim to show that our generated evaluation partners could approximate deployment-time partners. We admit that quantitative metrics and statistical analysis are reasonable methods for comparing behavior diversity, however, as in previous research, it is hard to get statistical results in measuring behavior diversity [5]. Thus, we present Figure 3 as a qualitative illustration, which can show not only the behavior diversity of different evaluation partners but also that previously used evaluation partners have similar behaviors and an intuitive illustration of how our generated evaluation partners cover possible deployment-time behaviors.
>
> Then, we want to point out that we validated the effectiveness of our method using various approaches from multiple dimensions, including the behavior diversity of evaluation partners shown in Figure 3 and the results from human experiments verifying that the evaluation results under our evaluation partners and the results under human players are highly similar in Table 2. Figure 3 is part of our effectiveness validation but not the whole. The lack of quantitative metrics and detailed statistical analysis in Figure 3 does not affect the full validation of the ZSC-Eval's effectiveness since we have already presented quantitative metrics and detailed statistical analysis in Table 2 to verify our ZSC-Eval's effectiveness.
>
> >Q: **Insufficient details are provided on the design and implementation of behavior-preferring rewards. Although Table 7 in the appendix lists events rewards chosen for the Overcooked scenarios, it does not explain how to decide which events to include and how to assign their weights. ..., clear guidelines on choosing appropriate events and their corresponding weights for the behavior-preferring rewards would be beneficial.**
>
> A: Behavior-preferring rewards consist of events in the environment and event weights. The events represent various situations during the deployment time. The event weights represent the partner's preference for the events. In our design, we enumerate all possible key actions for making soups in Overcooked as events to cover the possible preferences the deployment-time partners might have. And for Google Research Football, we enumerate all possible interactions with football as events. The event weights we selected are based on experiments. Specifically, we conduct enumerated experiments to find the suitable combination of weights that can help generate the most diverse evaluation partners.
>
> Guidelines for designing events and event weights, i.e., reward design guidelines, are difficult and are an inherent challenge in reinforcement learning. The design of behavior-preferring rewards requires the designer to have a comprehensive understanding of the environments and tasks. A promising further direction is to leverage some automatic reward design techniques, e.g., leveraging the large language models and high-level specification languages for reward generation [6].

---

> > ### Author Rebuttal · Authors · 2024-08-17
> >
> > >Q: **Evaluation should typically involve at least 10 seeds due to the non-deterministic behavior of RL algorithms.**
> >
> > A: Using more seeds for experiments indeed helps to reduce the non-deterministic of the experiments. To our knowledge, in the field of Multi-Agent Reinforcement Learning (MARL), due to the high computational resource demands of MARL environments, experiments typically use 5 seeds [7-9]. For example, most of the ZSC algorithms we are currently evaluating are tested using 5 seeds in their papers [10]. Due to computational resource constraints, we followed the usual practice in the evaluated algorithms' experiments and typical MARL research to use 5 seeds to conduct the experiments. Considering the time required for experiments, we plan to increase the number of random seeds in the future version of the paper to reduce the uncertainty of the experiments.
> >
> > >Q: **The computational resources used for the experiments are not specified.**
> >
> > A: Due to space limitations of the main text, we have already provided details of the computational resources used in the experiments in Appendix C.3 (Important Implementation Details) and mentioned in the main text as the experiment setup in Section 5. We have also provided the location of 'the total amount of compute and the type of resources used' description in the manuscript as required in the checklist.
> >
> > All our experiments were run on Linux servers including two types of nodes: 1) 1-GPU node with NVIDIA GeForce 3090Ti 24G as GPU and AMD EPYC 7H12 64-Core Processor as CPU, 2) 2-GPU node with two GeForce RTX 3090 24G as GPUs and AMD Ryzen Threadripper 3970X 32-Core Processor as CPU.
> >
> > >Q: **The toolkit has been tested in a limited number of agent scenarios, which could constrain the applicability of the findings across different environments that require coordination of many agents.**
> >
> > A: Principally, as formatted in Section 3.1, our ZSC-Eval can be extended to scenarios with a large number of agents. Empirically, in scenarios with a large number of agents, our ZSC-Eval framework can still generate evaluation partners as long as the approximate NEs can be reached by the underlining MARL algorithms, e.g., MAPPO [7]. In other words, the effectiveness of our framework is not affected by the number of agents. Therefore, our conclusions and findings are not constrained by the number of agents in the current environment.
> >
> > Furthermore, when training evaluation partners, the agents except those who interact with the ego agent frequently can be recognized as one agent to reduce the training complexity. Since adding more environments can benefit the research of zero-shot coordination, we are also happy to integrate environments and tasks with multiple agents in future research.
> >
> >
> > >Q: **An in-depth analysis of failure cases where the ego agent struggles to coordinate with partners could provide insights into the limitations of the proposed methods and help improve future iterations.**
> >
> > A: Thank you for the suggestion. We have already provided a failure analysis of some ZSC algorithms in Figure 16 of Appendix D.4. In detail, on the right side of Figure 16, we analyze why increasing population sizes in the Forced Coord. layout in Overcooked environment does not improve the performance by showing the BR-Diversity of different population sizes. Results show that an increase in population size does not lead to more diverse training agents, and thus does not improve the performance. We will discuss more failure cases in the future version of the main paper.
> >
> >
> > >Q: **Providing more empirical evidence would confirm metrics (Best-Response Diversity (BR-Div) and Best-Response Proximity (BR-Prox)) effectiveness and reliability.**
> >
> > A: For BR-Div, the behind intuition has been explained in Section 4.2 and Figure 2(a), i.e., an evaluation method should expose the ego agent to evaluation partners with diverse BRs, assessing the ZSC capability. We have further verified the effectiveness and reliability of BR-Div compared with other diversity metrics in Appendix D.2.
> >
> > For BR-Prox, the intuition is to balance the performance with evaluation partners of different cooperation capabilities, as explained from Line 227 to Line 234. The performance balancing technique is common in measuring solutions with multiple objectives [11].
> >
> > **We respectfully request that you consider our rebuttal and increase the score of our manuscript if your concerns have been adequately addressed.
> > If you have any further questions or require additional information, please let us know.**

---

> > > ### Author Rebuttal · Authors · 2024-08-17
> > >
> > > Reference:
> > >
> > > [1] Fu, Justin, et al. "Variational inverse control with events: A general framework for data-driven reward definition." *Advances in neural information processing systems* 31 (2018).
> > >
> > > [2] Brohan, Anthony, et al. "Do as i can, not as i say: Grounding language in robotic affordances." *Conference on robot learning*. PMLR, 2023.
> > >
> > > [3] Ye, Deheng, et al. "Mastering complex control in moba games with deep reinforcement learning." *Proceedings of the AAAI Conference on Artificial Intelligence*. Vol. 34. No. 04. 2020.
> > >
> > > [4] Shi, Zhenyu, et al. "Learning expensive coordination: An event-based deep RL approach."*International Conference on Learning Representations*. 2020.
> > >
> > > [5] Sarkar, Bidipta, Andy Shih, and Dorsa Sadigh. "Diverse conventions for human-AI collaboration." *Advances in Neural Information Processing Systems* 36 (2024).
> > >
> > > [6] Jothimurugan, Kishor. *Specification-Guided Reinforcement Learning*. Diss. University of Pennsylvania, 2023.
> > >
> > > [7] Yu, Chao, et al. "The surprising effectiveness of ppo in cooperative multi-agent games." A*dvances in Neural Information Processing Systems* 35 (2022): 24611-24624.
> > >
> > > [8] Rashid, Tabish, et al. "Monotonic value function factorisation for deep multi-agent reinforcement learning." *Journal of Machine Learning Research* 21.178 (2020): 1-51.
> > >
> > > [9] Carroll, Micah, et al. "On the utility of learning about humans for human-ai coordination." *Advances in neural information processing systems* 32 (2019).
> > >
> > > [10] Strouse, D. J., et al. "Collaborating with humans without human data." *Advances in Neural Information Processing Systems* 34 (2021): 14502-14515.
> > >
> > > [11] Zhang, Yifan, et al. "Deep long-tailed learning: A survey." *IEEE Transactions on Pattern Analysis and Machine Intelligence* 45.9 (2023): 10795-10816.

---

> > > > ### Comment · Reviewer_jqjf · 2024-08-19
> > > >
> > > > Thank you for your detailed and thorough rebuttal. I appreciate the clarifications provided, particularly the references to specific sections in the appendix that address my concerns, such as the failure case analysis in Appendix D.4 and the validation of BR-Div and BR-Prox in Appendix D.2. I am satisfied with your explanations and will increase my score by 1, as your responses have convincingly addressed my points. However, I strongly encourage the authors to continue developing the code to support additional environments and consider increasing the number of seeds in future experiments, as recent MARL literature suggests the use of 10 seeds for more robust evaluations [1]. I understand that ZSC experiments are computationally intensive, so this recommendation may need to be adapted accordingly.
> > > >
> > > >
> > > > [1] Gorsane, Rihab, et al. "Towards a standardised performance evaluation protocol for cooperative MARL." Advances in Neural Information Processing Systems 35 (2022)

---

> > > > > ### Author Rebuttal · Authors · 2024-08-19
> > > > >
> > > > > Thank you very much for your positive feedback and for increasing your score. We appreciate your suggestion regarding supporting additional environments and increasing the number of seeds in future experiments. We recognize the importance of these aspects in MARL research and will consider them in our future work.
> > > > >
> > > > > Best regards,
> > > > >
> > > > > Authors of Submission 765

---

### Official Review · Reviewer_yyn1 · 2024-07-25
**Nice work but there is potential for advancement**

**Rating:** 7
**Confidence:** 3
**Correctness:** Appears correct
**Clarity:** Clear

**Review:**

The quality and clarity of the paper is good. The Presentation is also good.

pros:
1. The paper addresses an important challenge in MARL - evaluating ZSC capabilities. The problem of zero-shot coordination is increasingly relevant as AI agents need to work with diverse partners without prior coordination.
2. The three components of ZSC-Eval are well-designed and address the limitations of current evaluation methods.
3. ZSC-Eval provides a comprehensive evaluation toolkit and benchmark for ZSC algorithms. The benchmark results also provide insights into the performance and limitations of current ZSC algorithms.

cons:

It would be beneficial to provide the benchmark results on more tasks. And more in-depth discussion of the limitations would be better.

**Strengths:**

1. The paper addresses an important challenge in MARL - evaluating ZSC capabilities. The problem of zero-shot coordination is increasingly relevant as AI agents need to work with diverse partners without prior coordination.
2. The three components of ZSC-Eval are well-designed and address the limitations of current evaluation methods.
3. ZSC-Eval provides a comprehensive evaluation toolkit and benchmark for ZSC algorithms. The benchmark results also provide insights into the performance and limitations of current ZSC algorithms.

**Additional Feedback:**

N/A

**Documentation:**

Yes, the GitHub repository is detailed and provides the necessary details for implementation

**Limitations:**

The authors could consider discussing the potential limitations of this work in more detail.

**Opportunities For Improvement:**

1. Figure 2(a) is missing.
2. It would be beneficial to provide the benchmark results on more tasks in Google Research Football.
3. The paper could benefit from a more in-depth discussion of the limitations of ZSC-Eval.

**Relation To Prior Work:**

Ample citations and prior discussion

**Summary And Contributions:**

The paper presents ZSC-Eval, an evaluation toolkit and benchmark for Zero-shot Coordination (ZSC) algorithms in cooperative multi-agent reinforcement learning (MARL). The paper addresses the challenges of evaluating ZSC capabilities, including the mismatch between evaluation and deployment-time partners and the inadequacy of current evaluation metrics.  ZSC-Eval consists of three components: generation of behavior-preferring agents as evaluation partner candidates, selection of evaluation partners based on best response diversity, and measurement of ZSC capability using the best response proximity metric. The paper provide empirical findings about the effectiveness of ZSC-Eval in generating diverse evaluation partners, the consistency between ZSC-Eval results and human evaluation, and the performance of current ZSC algorithms in different environments.

---

> ### Author Rebuttal · Authors · 2024-08-16
>
> We thank the reviewer for the insightful suggestions and recognition of our work.
> For your suggestions and problems, we reply point by point:
>
> > Q: **Figure 2(a) is missing.**
>
> A: Figure 2(a) illustrates that different partners may respond to similar BRs.
>
> Regarding the missing problem with Figure 2(a), we have checked with multiple different system devices including Linux, MacOS and Windows, and did not find any issues with Figure 2(a) being missing in the PDF we submitted. We have attached Figure 2 in PDF format in this rebuttal for reference.
>
> We are not sure if your comment about 'Figure 2(a) is missing' refers to Figure 2(a) being invisible in the submitted PDF; if not or if you have any other problems with Figure 2(a), please let us know.
>
> > Q: **It would be beneficial to provide the benchmark results on more tasks in Google Research Football.**
>
> A: Our work is a framework including training, and evaluating ZSC algorithms among different tasks. We are happy to integrate more environments and tasks to further promote the research of zero-shot coordination. We will continuously develop our code repository to include more algorithms, environments and tasks and present more empirical results in further discussion if time permits.
>
> >Q: **The paper could benefit from a more in-depth discussion of the limitations of ZSC-Eval.**
>
> A: The limitations of our work mainly lie in that the design of event-based rewards needs careful handcrafts and that event-based rewards may not fully represent deployment-time partners. The events represent various situations during the deployment time, which requires the designer to have a comprehensive understanding of the environments and tasks, and is hard to exhaust. The limitations result from the challenge of reward design, which is an inherent challenge in reinforcement learning. To alleviate these limitations, a promising further direction is to leverage some automatic reward design techniques, e.g., leveraging the large language models and high-level specification languages for reward generation [1]. We will discuss the limitations in the further version of our paper.
>
>
> **We respectfully request that you consider our rebuttal and increase the score of our manuscript if your concerns have been adequately addressed.
> If you have any further questions or require additional information, please let us know.**
>
> Reference:
>
> [1] Jothimurugan, Kishor. *Specification-Guided Reinforcement Learning*. Diss. University of Pennsylvania, 2023.

---

> ### Author Response · Authors · 2024-08-22
> **A Gentle Reminder for Rebuttal Consideration**
>
> Dear Reviewer yyn1,
>
> Thanks again for your valuable comments and suggestions. We have submitted our rebuttals to the reviews. We explain the problems and try our best to settle your concerns. We hope that your concerns are well addressed, e.g., whether we have explained Figure 2(a) clearly.
>
> We look forward to your response and are eager to continue our discussion.
>
> Sincerely, Authors

---

### Author Rebuttal · Authors · 2024-08-16

Dear reviewers,

We thank all reviewers for the insightful suggestions and appreciate all reviewers' praise and recognition of our work.

We hope our responses below have addressed your concerns.

Our work makes unique contributions to the zero-shot coordination problem as an evaluation tool and a benchmark, including the fact that we are the first to investigate the evaluation of ZSC capability and generate evaluation partners with awareness of deployment-time requirements. We believe these contributions are of significant relevance to research in this field.

If you have any further questions, please don't hesitate to engage in discussion with us, and we will respond promptly.

Sincerely,

Authors

---

### Decision · Program_Chairs · 2024-09-26

**Decision:**

Accept (Poster)

**Comment:**

This paper proposes a toolkit and benchmark for evaluating Zero-Shot Coordination (ZSC) in Multi-Agent Reinforcement Learning (MARL). The setting of ZSC involves an 'ego'-agent that can be deployed to coordinate with unseen 'evaluation' partners without additional training. The reviewers have acknowledge that the proposed ZSC-Eval framework is well-designed and comprehensive (yyn1), while being grounded in strong theoretical principles and experimental validation (jqjf). Despite describing the evaluation as being comprehensive, multiple reviewers also pointed out concerns over limited generalizability of findings given the limited diversity of environments (Overcooked, Google Research Football) and limited scenarios/tasks within these environments.

Limited scenarios/environments notwithstanding, I found the paper to be well-written and thorough and concur with the reviewers' opinions of the strengths of the work and its potential for advancing this crucial research space. The authors' rebuttal indeed addressed the concerns of the reviewers resulting in raised scores.

I find it important enough to mention that it is a bit self-contradictory that in the rebuttal the authors acknowledge limitations arising from reward design--"an inherent challenge in reinforcement learning"--and simultaneously assert that the limitation is relatively simple to address. In the response to yyn1, the authors acknowledge that the central limitations arise from the design of event-based rewards, "which requires the designer to have a comprehensive understanding of the environments and tasks, and is hard to exhaust".  Meanwhile, in the response to jqjf, the authors state that indeed the key to generalization is the design of these event-based rewards, which are "relatively simple to implement in a well-studied environment where some reward and feature engineering has been completed".

I suggest that the authors make up their mind. I tend to agree with the first of the two stances, and a straight-forward acknowledgment of the limitations and a discussion about how future works might address the limitations will make the paper stronger in my opinion. The latter reframing comes across as a bit of a disingenuous acknowledgment of the reviewers' concerns, and detracts from an otherwise well-argued rebuttal. I encourage the authors to address this in the final version of the paper.

Nevertheless, given the overall quality, the importance of the work in advancing an important research question within the MARL community, and the expected continued expansion by the authors of the "code repository to include more algorithms, environments and tasks", I am recommending including this paper within the program this year.